# The Critical Raw Materials in Cutting Tools for Machining Applications: A Review

**DOI:** 10.3390/ma13061377

**Published:** 2020-03-18

**Authors:** Antonella Rizzo, Saurav Goel, Maria Luisa Grilli, Roberto Iglesias, Lucyna Jaworska, Vjaceslavs Lapkovskis, Pavel Novak, Bogdan O. Postolnyi, Daniele Valerini

**Affiliations:** 1ENEA–Italian National Agency for New Technologies, Energy and Sustainable Economic Development, Brindisi Research Centre, S.S. 7 Appia–km 706, 72100 Brindisi, Italy; daniele.valerini@enea.it; 2School of Engineering, London South Bank University, 103 Borough Road, London SE1 0AA, UK; saurav.goel@cranfield.ac.uk; 3School of Aerospace, Transport and Manufacturing, Cranfield University, Cranfield MK4 30AL, UK; 4ENEA–Italian National Agency for New Technologies, Energy and Sustainable Economic Development, Casaccia Research Centre, Via Anguillarese 301, 00123 Rome, Italy; marialuisa.grilli@enea.it; 5Department of Physics, University of Oviedo, Federico Garcia Lorca 18, ES-33007 Oviedo, Spain; roberto@uniovi.es; 6Łukasiewicz Research Network, Institute of Advanced Manufacturing Technology, 30-011 Krakow, Poland; lucyna.jaworska@ios.krakow.pl; 7Faculty of Non-Ferrous Metals, AGH University of Science and Technology, 30-059 Krakow, Poland; 8Faculty of Civil Engineering, Scientific Laboratory of Powder Materials/Faculty of Mechanical Engineering, Institute of Aeronautics, 6A Kipsalas str, lab. 110, LV-1048 Riga, Latvia; lap911@latnet.lv; 9Department of Metals and Corrosion Engineering, University of Chemistry and Technology, Prague, Technická 5, 166 28 Prague 6, Czech Republic; Paja.Novak@vscht.cz; 10IFIMUP—Institute of Physics for Advanced Materials, Nanotechnology and Photonics, Department of Physics and Astronomy, Faculty of Sciences of the University of Porto, 687 Rua do Campo Alegre, 4169-007 Porto, Portugal; b.postolnyi@gmail.com; 11Department of Nanoelectronics, Sumy State University, 2 Rymskogo-Korsakova st., 40007 Sumy, Ukraine

**Keywords:** critical raw materials, cutting tools, new materials, new machining methods, modelling and simulation

## Abstract

A variety of cutting tool materials are used for the contact mode mechanical machining of components under extreme conditions of stress, temperature and/or corrosion, including operations such as drilling, milling turning and so on. These demanding conditions impose a seriously high strain rate (an order of magnitude higher than forming), and this limits the useful life of cutting tools, especially single-point cutting tools. Tungsten carbide is the most popularly used cutting tool material, and unfortunately its main ingredients of W and Co are at high risk in terms of material supply and are listed among critical raw materials (CRMs) for EU, for which sustainable use should be addressed. This paper highlights the evolution and the trend of use of CRMs) in cutting tools for mechanical machining through a timely review. The focus of this review and its motivation was driven by the four following themes: (i) the discussion of newly emerging hybrid machining processes offering performance enhancements and longevity in terms of tool life (laser and cryogenic incorporation); (ii) the development and synthesis of new CRM substitutes to minimise the use of tungsten; (iii) the improvement of the recycling of worn tools; and (iv) the accelerated use of modelling and simulation to design long-lasting tools in the Industry-4.0 framework, circular economy and cyber secure manufacturing. It may be noted that the scope of this paper is not to represent a completely exhaustive document concerning cutting tools for mechanical processing, but to raise awareness and pave the way for innovative thinking on the use of critical materials in mechanical processing tools with the aim of developing smart, timely control strategies and mitigation measures to suppress the use of CRMs.

## 1. Introduction

In the era of globalisation and high competitiveness, it is of utmost importance for industries to work on reducing manufacturing costs and simultaneously providing added value in terms of increased life of the product when put into service. It is also imperative for them to have a flawless supply chain which contains a key component of sourcing critical raw materials (CRMs) [1] for them to remain sustainable.

In the EU, the excess of imports of CRMs, especially in the mechanical manufacturing industry—here, we refer to metals that are vital to EU industries such as tungsten, chromium, and niobium to name but a few—has reached an alarming stage, bringing a great degree of dependency on countries that are monopolistic suppliers of these CRMs. To mitigate this foreseeable problem, the EU has already launched several campaigns, including the “Raw Materials Initiative” (RMI) [2]. 

The issue of CRMs must be tackled with scientific rigour by pursuing different parallel actions; in particular, by (1) improving the production processes of CRMs (increasing sustainable mining, reducing extraction costs, increasing the efficiency of materials, increasing security, etc.); (2) finding suitable candidates to partially or totally substitute the CRMs; and (3) increasing their recycling.

In this context, the development of new materials with superior characteristics or better performance than existing materials is desirable to lead to a longer product life and therefore to reduce the cost of the product. The cost of cutting tools used in the manufacturing industry at present, particularly to machine high-value components such as turbine blades, automotive and aerospace parts, machine parts and biomedical implants, is significant and dictates the total manufacturing cost and thus the final price of the product [3]. The tooling cost is not only associated with the cost incurred on relapping the tools, but it also involves the cost incurred due to the increased cycle time owing to the unloading and reloading of the new tool. The main focus in manufacturing research currently is to improve productivity, and thus, there is a focus on developing materials (especially for cutting tools) that can withstand higher cutting loads than the current attainable machining limits. The cutting tools discussed here are meant for all mechanical-based contact loading processes, including, for example, milling, drilling, turning, honing, chamfering, hobbing, knurling, parting-off, and so on. The current development of new materials specifically for cutting tools has remained a slow process, particularly due to the high initial investment, but interest has been growing recently towards developing more sustainable ways to de-risk the supply chain. 

Cemented carbide is one of the most popular cutting tool materials. Typically, a carbide cutting tool is manufactured with a mixture of tungsten and cobalt (the binder that holds tungsten carbide together), with a wealth of variations in carbide grain size and the ratio of carbide to binder. Preferred blends have been developed over time to achieve effectiveness with different cutting depths and widths, as well as workpiece materials. The distribution of cutting tools in the global market in 2018 by the cutting method and workpiece material is shown in Figure 1a,b, respectively. It can be seen that milling, turning and drilling are the processes mostly used in primary machining operations, and the tools for these operations capture almost 87% of the total tooling market. In terms of the type of cutting material, cemented carbides capture one half of the market, followed by high speed steel. Ceramics, cermets and superhard materials such as polycrystalline diamond (PCD) and polycrystalline cubic boron nitride (PCBN) capture the remaining share of the tool material market. The two important ingredients of the carbide tools mentioned above—namely tungsten and cobalt—were earlier identified (in 2011) as being among the list of the 14 critical raw materials (see Figure 2) vital to EU industries [4]. More recently, in 2014 and in 2017, they have continued to remain on this list on account of their economic importance and risk of supply interruption [5,6]. Another driving force for Co substitution lies in its well-known genotoxic and carcinogenic activities. A pressing need has thus emerged for the EU to develop alternative solutions to the existing cutting tool materials to avoid a complete closure of machining businesses in the absence of the raw material of the cutting tools.

Perhaps, the most direct impact of the new era of digital manufacturing [8] on future cutting tools is the potential use of additive manufacturing technologies for the generation of quasi-network-shaped tools with geometric structures and challenging sub-structures and the potential to produce under-functionally different structures and optional materials with graded properties (4D printing). One of the four themes defined by the Roadmap of High Performance Cutting (HPC) [9] points at the use of integrated tooling based on cycle time reduction and quality improvement in high-end batch production. This theme was further developed, and the term “smart tooling” has been coined [9], which refers to the insertion of sensor actuators [10] within the cutting tool as a step forward in the process monitoring of the tool, allowing users to collect the necessary data for the creation of more accurate digital twins for the machining processes (Figure 3). 

An example of the additional functions which are commercially available in this regard is the use of low-frequency vibration-assisted drilling (LFVAD) [11] for improving the quality of the machining of brittle materials such as carbon fibre reinforced composite (CFRP) via an interrupted cutting mechanism.

Besides such alterations in machining mechanisms, there is remaining interest regarding the successful realisation of novel tool coating materials as well as the development of enhanced cutting technologies, including laser-assisted cutting and cutting with cryogenic assistance. The idea behind the use of these methods depends on whether the main wear mechanism of the tool is governed by the high physical hardness of the workpiece or the high chemical affinity. In the former case, laser heating ahead of the cutting tool can reduce the cutting resistance, making the workpiece more compliant to cutting, while in the latter case, the cryogenic environment helps delay the kinetics of the chemical reactions, thus inhibiting the occurrence of diffusion-induced chemical wear. 

As an additional strategy, there are many existing coatings that protect cutting tools working under extreme conditions. The most desired attribute of tool coatings is to provide high hardness and toughness for high wear performance and thermal and chemical stability to withstand extreme cutting environments, especially during high-speed machining. Most of the widely used coatings do not contain CRMs, but multi-elemental and high-entropy coatings designed for specific extreme conditions may include them [12]. Transition metals are one of the biggest groups of chemical elements in the periodic table and, as can be seen in Figure 2, half of the refractory metals group are marked as CRMs. However, the amount of material needed for a coating is significantly lower than for analogous bulk CRMs used in cutting tools. Moreover, the nano and microstructural design of protective coatings, such as nanocomposites or multilayers, suppresses the usage of certain CRMs in coatings. 

Several recent studies have addressed technical–scientific actions with different methods that can be pursued to address the problems of CRMs in cutting tools; in particular, to achieve the following:Increase the life of the tools by enhanced material removal techniques including laser assistance, cryogenic assistance, vibration assistance and use of protective coatings;Develop and synthesise newer materials that can adequately partially or totally replace the CRMs used in the tools;Rigorously involve modelling and simulation in the tool design and deploy digital twins to make improved predictions in the present era of digital manufacturing;Improve the recycling of worn-out tools.

In this paper, we summarise the current understanding regarding the use of CRMs in the tooling industry, considering the four aspects listed above, without any intentional effort to propose an immediate solution to this problem, but with the aim of stimulating the development of effective strategies in this field.

## 2. Attempts to Expand the Life of Tungsten Carbide (WC)–Co Based Tools

There are several methods to increase the lifetime and efficiency of tools in machining applications, such as the proper modification of the base tungsten carbide (WC)–Co material of the tool, the use of advanced processing techniques, and the application of protective coatings. These methods are reviewed in the following subparagraphs.

### 2.1. WC–Co Based Materials

Cemented carbides belong to the most common and the longest-used tool materials produced by powder metallurgy methods. Sintered carbides are characterized by their high strength and abrasion resistance and include one or more high-melting metal carbides constituting the basic component together with the metallic binding phase [13]. According to this standard, sintered carbides are divided into groups: those used for the production of tools for metal machining, plastic forming and for the use of mining tools. The basic component of cemented carbides is WC, which, depending on the manufacturer and group of material applications, can constitute from 50% to 90% by weight of the sintered content. The other ingredients are carbides of titanium (TiC), tantalum (TaC) and niobium (NbC), the content of which can be from 0% to 35% by weight. These carbides dissolve each other and can also dissolve a large amount of tungsten carbide. The rest of the composition is usually cobalt. Cobalt is characterized by very good wettability with most materials that are components of sintered tool materials and a fairly high melting point, which is 1493 °C. Such a microstructure, which is characteristic for sintered carbides, allows for the presence of a ductile binding phase; additionally, the hard and brittle carbide phase allows the bonding of opposing features in one material, such as high abrasion resistance and hardness with high strength and fairly good ductility. Many carbide applications feature very good abrasion resistance, which depends on the chemical composition. 

Two-component cemented carbides of the WC–Co type with a low content of cobalt are characterized by the highest abrasion resistance. These grades can be used if there are no impacts during operation, and abrasion is the main wear mechanism.

The universality of WC–Co results from its very good mechanical and tribological properties. The technological process of obtaining cemented carbides is characterized by relatively easy formation, sintering temperatures lower than ceramic materials and electrical conductivity, and these factors have a positive effect on the possibility of shaping products with complicated geometry using erosive treatment as well as on the ease of applying anti-wear and anti-corrosion coatings. The disadvantage of this material is that cutting tools made of cemented carbides work at relatively low cutting speeds, and these materials tend to oxidize already at temperatures above 400 °C. The most commonly used carbides are tungsten, titanium, tantalum or other high-melting metal elements in an amount of 75 to 94 wt.% and cobalt, nickel or molybdenum, and sometimes other metals are usually used as the binding matrix. The toughness of WC–Co can be increased by increasing the cobalt content, whilst the wear resistance is increased by decreasing the cobalt content and decreasing the carbide grain size. The substitution of cubic carbides (TiC, NbC and TaC) for WC leads to improvements in wear resistance and resistance to plastic deformation. The microstructure of cemented carbide WC–Co consists of tungsten carbide particles combined with cobalt, which are obtained in the sintering process with the participation of the liquid phase. When machining a metal with high plasticity, such as pure iron, using conventional WC–Co cemented carbide, a chip tends to adhere on the rake face of the cutting tool, resulting in serious adhesive wear due to the existence of cobalt, which has a lower melting point compared to WC. The presence of cobalt in cemented carbides during the machining of steel, for example, causes the chip to stick to the cutting blade. Crater wear is caused by the chemical interaction between the rake face and hot chip. Wear occurs by the diffusion of the tool material into the chip or by the adhesion between the chip and tool followed by a fracture below the adhered interface within the tool material. Crater wear can be reduced by increasing the chemical stability of the tool material, decreasing the solubility of the workpiece or barrier protection by substrate alloying or coating [14]. Thermal shock cracks are caused by large temperature gradients at the cutting edge. There is a large difference in the coefficients of thermal expansion between cobalt and tungsten carbide, which is why cracks may appear on the blade during its operation. Therefore, cemented carbides are often used as substrates for coatings.

However, studies are being carried out on uncoated WC–Co intended for machining materials. An example consistent with the idea of reducing the use of critical materials in cemented carbides is cobalt-free carbide. The first cobalt-free carbides were obtained by conventional methods from micropowders [15]. These materials were characterized by high hardness and abrasion resistance, but exhibited greater brittleness compared to cemented carbide WC–Co. For binderless tungsten carbide, the sintering took place in a solid phase, which for conventional, pressure-free sintering means higher sintering temperatures and/or longer sintering times, which in turn cause grain growth in the polycrystalline material. The use of nano-powders and pressure sintering methods (spark plasma sintering, hot pressing sintering) improved the properties of the cobalt-free cemented carbides [16,17,18,19].

The addition of free carbon to the binderless WC reduces the amount of the brittle phase of W_2_C, Co_3_W_3_C and oxide creation [20,21,22]. 

Spark plasma sintering (SPS, also called “Field Assisted Sintering Technology”) and hot pressing sintering, thanks to the shorter sintering time and pressure, allow the reduction of grain growth and of the porosity of materials sintered in the solid phase [23].

Composite materials with the addition of ceramic powders are another idea which has been developed in relation to cemented carbides. In the case of ZrO_2_, additional reinforcement is used, which is the result of stresses created during the ZrO_2_ phase transformation. They are usually added in an amount of 5% to 15 wt.% in the form of oxides, carbides and nitrides [24,25]. 

This idea is consistent with the substitution of the critical materials of W and Co. Composite materials with a WC or WC–Co matrix have recently been developed very intensively thanks to the SPS method [26]. There are already individual examples of the industrial application of such solutions. 

Research related to the reduction of cemented carbide consumption is also associated with attempts to increase their durability. Superhard materials such as diamond and cubic boron nitride (cBN) are introduced into the WC–Co [27]. Composites containing ultrafine tungsten carbide/cobalt (WC–Co) cemented carbides and 30 vol.% cBN were fabricated mainly by FAST (field-assisted sintering technology) methods. WC–Co/cBN composites have been considered as a next-generation material for use in cutting-tool edges and are characterized by an optimal combination of hardness and toughness. The major challenge in sintering these composites is to produce a well-bonded interface between the WC–Co matrix and cBN particles [28,29,30,31].

Whisker toughening is mainly used for binderless tungsten carbide. There are known studies in which SiC_w_, Si_3_N_4w_ and Al_2_O_3_ whiskers were used. The whisker participation was up to 10 vol.% [32,33].

The literature describes research related to toughening using nanotubes and graphene to improve the thermal conductivity of cemented carbides. The study found that thermal stress is the main reason for the failure of cemented carbide shield tunnelling tools when shield tunnelling is carried out in uneven soft and hard soil [34].

### 2.2. Advanced Machining Techniques

The machining of “difficult to cut materials” such as the nickel-based superalloys used in turbine blades requires an extended tool life for unhindered machining [35]. Traditional carbide tools are limited to working in the range of 30 m/min to 70 m/min because of their poor thermochemical stability; however, they can be used at high feeds due to their high toughness. Improvements in the poor thermochemical stability of cemented carbide tools could be aided by advanced modelling. Additive manufacturing technology [36] and new machining techniques have now opened newer possibilities of making complex tool shapes. This section presents a brief review of recent attempts made to expand tool life either by reducing the load on the cutting tool or by deferring the known pathway of the accelerated wear mechanism. In either case, the end result is the improved quality of the machined part.

#### 2.2.1. Machining by Laser Assistance (Thermal-Assisted Machining)

In pursuit of overcoming the challenges of difficult-to-machine materials, laser assistance has been adopted worldwide for incorporation with mechanical micromachining. To date, seven major patents (in the years 1982 [37,38], 2006 [39], 2013 [40], 2011 [41], 2014 [42], 2016 [43], and 2017 [44]) have been granted in the US concerning the use of laser assistance during mechanical micromachining. All these patents can be connected by a single chain of a process which is now well known as thermal-assisted machining (TAM) [45]. The concept of TAM is schematically shown in Figure 4a and relies on pre-heating the cutting zone of the material being cut ahead of the cutting tool. This methodology reduces the physical hardness of the workpiece, making it more compliant to cutting by reducing the specific cutting energy (i.e., the work done by the tool in removing a unit volume of the material).

#### 2.2.2. Cryogenic Machining

As shown in Figure 4b, cryogenic machining relies on freezing the cutting tool to extreme temperatures of about −196 °C by means of either liquid nitrogen or liquid carbon dioxide [49]. Cryogenic machining helps to serve two purposes: (i) it elevates the relative hardness gradient between the tool and the workpiece; and (ii) it delays the kinetics of any chemical diffusion that may likely trigger the tribo-chemical wear of the tool. S. Rakesh highlighted how cryogenic technology is ecological, non-toxic and non-explosive [50].

Cryogenic cooling has been executed in cutting operations in different ways by using liquid nitrogen for precooling the workpiece, cooling the chip, and cooling the cutting tool and cutting zone [51]. Numerous studies have compared conventional cutting strategies and cryogenic cooling methods. However, studies conducted in an attempt to determine the best technique found many contradictions, as the conclusions described above could change in relation to tool–work pairs, to the cutting conditions and to the general evaluation parameters.

The literature dealing with tungsten carbide cutting tools in cryogenic processing is significantly less numerous compared to the literature on steel tools, and only a few papers consider an evaluation parameter of the tool’s lifetime [52]. In this context, all these studies show that the cryogenic treatment has positive effects both on the lifetime of the tools and on the surface finishing of the product. The cryogenic treatment of the materials showed significant positive effects such as increased wear resistance, reduction of residual stresses, increased hardness and fatigue resistance.

Seah et al. [53] carried out a series of experiments with the aim to study the aspects of cold and cryogenic treatments on uncoated WC inserts for carbon steel ASSAB 760. They showed that at different cutting speeds the cryo-treated inserts exhibited greater resistance to wear than the untreated and recovered counterpart. In addition, they found that the cold and cryogenic treatment significantly increased the resistance of the cutting insert to chip removal, which became increasingly important as the cutting speed increased. 

Yong et al. [54] subjected uncoated WC inserts to a cooling treatment down to −184.5 °C for 24 h and then heated the insert to room temperature, keeping the rate of 0.28 °C/min unchanged both when increasing and decreasing the temperature. They developed a series of face milling operations using different cutting speeds but kept all the other processing parameters constant using untreated and cryo-processed inserts. Two pieces of information were highlighted: the first concerned the cryo-processed inserts, which generally performed better than their untreated counterparts; and the second focused on the increase in tool life of 28%–38% during the cryogenic treatment in wet machining compared to dry machining. 

Sreeramareddy et al. [55] studied the tool wear, cutting forces and the surface finish of parts worked using a WC insert coated with a multilayer and subjected to cryogenic treatment. They showed that cryogenic processing reduced the wear on the side of the inserts as well as the cutting forces and the surface roughness of the workpiece machined with untreated inserts.

These studies suggest that the cryogenic treatment of carbide tools is capable of improving the productivity and quality of the final product because it guarantees greater resistance to wear and surface finish. Improvements have been reported in the red-hardness of cryogenically treated inserts, which resulted in low flank wear [56].

Bryson [57] claimed that the increase in wear resistance, with a consequent increase in the lifetime of the carbide tool, was due to the greater strength of the binder after cryogenic treatment.

Thakur et al. [58] highlighted that WC tools undergo a less-strong microstructural modification under cryogenic treatment compared to that detected with conventional heat treatments; some physical transformations actually occur concerning the densification of cobalt, which induces an increase in the gripping of carbide particles and an improvement in the tool life amounting to a wear resistance increase of 27%. The cryogenic treatment of the tool is one approach to enhance its properties by introducing microstructural changes. The formation of complex compounds such as Co_6_W_6_C or Co_3_W_3_C might have increased the hardness in the samples due to forced air cooling and oil quenching.

In another work, the effect of cryogenic treatment on the tool was shown to include an increase in tool life, lower cutting force and better surface finish compared to the untreated condition [59].

Reddy et al. [60] examined the workability of C45 steel with untreated and treated (−110 °C for 24 h) ISO P-30 tungsten carbide inserts by measuring the flank wear, main cutting force and surface finish. They concluded that the best machinability found was caused by the increase of the thermal conductivity of tungsten carbide induced by cryogenic treatment. The surface roughness of the workpiece was lower by approximately 20% when the workpiece was machined with deep cryogenically treated tungsten carbide tool inserts in comparison with untreated inserts for cutting speeds in the range between 200 and 350 m/min. Vadivel et al. [61] studied the microstructure of cryogenically treated (TiCN + Al_2_O_3_) coated and untreated inserts in turning nodular cast iron. Their results highlighted that coated and treated tools have better properties that help the cutting tool to operate under hostile conditions for a longer time. 

A. Swamini et al. [62] drew up an overview of the metallurgy behind the cryogenic treatment of cutting tools; their results regarding WC–Co tools can be summarized as follows:Cryogenic treatment induces a structural variation with the formation of carbides of the eta phase and the redistribution or densification of Co, which increases its hardness;The micro-hardness of the treated tools is greater than that of untreated tools;The cryogenic treatment of tungsten carbide inserts increases the tool’s lifetime during the processing of austenitic AISI 316 steel;Wear patterns are smoother and more regular;Cryogenic treatment increases chipping resistance;The radius of the chip coil as well as the thickness of the chip itself is smaller after processing with cryogenically treated inserts or in cryo-processing conditions.

M.I. Ahmed et al. [63] used a modified tool holder for the efficient use of cryogenic cooling for machine cutting. The modified tool holder uses the direct continuous contact of liquid nitrogen with the cutting insert for the perfect cooling of the cutting tool. The results showed an average 30-fold increase in the lifetime of the carbide tools. 

M. Dhananchezian [64] described the machinability characteristics of turning Hastelloy C-276 with a Physical Vapour Deposition (PVD) coated nano-multilayer TiAlN carbide cutting tool using dry turning and liquid nitrogen cooling methods. The improvement of cutting tool performance was achieved under liquid nitrogen cooling by the control of the wear mechanisms, which in turn reduced the wear rate. Yildirim [65] examined the effect of some cooling conditions on machinability which may be an alternative to conventional cooling. The results showed that the 0.5 vol.% hBN cooling method used in conjunction with liquid nitrogen gave the best results in terms of the machining performance and lifetime of carbide tools.

Particularly noteworthy is the recent work of Biswal et al. [66] which, based on experimental results, highlighted how uncoated cryogenically treated tempered cermet inserts perform better than other cermet inserts thanks to their better wear resistance, micro-hardness and toughness. Despite all this, the cryogenic processing technique has not yet displaced conventional industrial processing. Further studies on cryogenic processing are necessary in order to highlight the increase in the lifetime of WC tools with the consequent saving of critical materials such as W and Co.

#### 2.2.3. Vibration-Assisted Machining

The literature refers to vibration-assisted machining (VAM) as the process of intermittent cutting. The reported benefits include the low wear rate of the tools, reduced burr formation on the workpiece and higher cutting depths being achieved. 

The attainable cutting speeds during VAM are limited by the hardware and system, and hence ultrasonic-assisted machining methods are classed as low-speed machining techniques [67]. Also, the process of vibration assistance can be implemented in two ways:
(i)uniaxial tool movement (1D VAM), where the tool vibrates in a plane parallel to the surface of the workpiece; and (ii)elliptical tool movement [68] (EVAM) where the tool vibrates with an elliptical motion. Both methods can be both resonance-based and non-resonance-based.

The resonant system operates at discrete frequencies, normally higher than 20 kHz and at amplitudes of less than 6 µm, whereas the non-resonant system operates at frequencies between 1 to 40 kHz and with amplitudes 10 times higher than the resonant system. As shown in Figure 4c, the tool is prescribed an oscillatory motion by a wiggle function or the tool is vibrated at a high frequency. Some of the reported benefits are highlighted in Table 1.

#### 2.2.4. Surface Defect Machining

It has been demonstrated that the surface defect machining (SDM) method harnesses the combined advantages of both porosity machining [74] and pulse laser pre-treatment machining [75], as shown in Figure 5a, by machining a workpiece initially by generating surface defects at depths less than the uncut chip thickness through either mechanical means and/or thermal means followed by a routine conventional machining operation. SDM enables ease of deformation by shearing the material at a reduced input energy [76,77]. Also, due to the large proportion of stress concentration in the cutting zone—rather than the sub-surface—a reduction in the associated residual stresses on the machined surface is enabled.

Figure 5b shows the provision of the structured surfaces on the rake face of the cutting tool. It has been implied particularly that when these structured surfaces are fabricated at a direction of 90° to the cut surface, they are helpful in reducing the extent of friction at the tool–chip interface. In contrast, probing these structures along the direction of cutting is rather disadvantageous as it destroys the integrity of the edge of the tool. Overall, a summary of these recent developments in this area of machining, as an add-on to the existing machines to improve the workpiece’s machinability, is summarised in Table 2.

### 2.3. Protective Coatings

Several types of failure mechanisms, such as delamination, abrasion, oxidation, diffusion, etc., result from the surface of the cutting tool. Failure occurs due to the interactions at the interface between tool and workpiece or tool and ambient medium, respectively. This leads to the conclusion that it is possible to protect a tool by surface treatment, creating some additional interface or coating with individually designed features. Figure 6 shows the distribution of uncoated and coated cemented carbide cutting tools, where the growth of the latter is clearly visible in a timeline perspective.

Protective coatings may significantly increase the lifetime of tools and thus reduce the consumption of CRM content in bulk materials. Furthermore, most modern coatings are CRM-free. The evolution of the protective effect of coatings by the extension of tools’ lifetimes and a comparison of coatings with available data from selected published works are shown in Figure 7. It is seen that coated cutting tools may have an extra lifetime of 200%–500% or more at the same cutting velocities. This can also lead to an increase in operational velocities (by 50% to 150%) with the same lifetime of cutting tools [93]. Coatings improve the endurance-related properties, and eventual wear failures may be dealt with through the wise application of the techniques outlined in Section 4.

Each application area requires specific properties of protective coatings to reach maximum performance. The extreme conditions of service of cutting tools mean that working parts face high wear, high pressure, elevated temperatures caused by high machining speeds, oxidation, and corrosion from lubricants or cooling agents. The main criteria of the evaluation of protective coatings are summarised in Figure 8, even though a compromise among the different prerequisites must be achieved. Usually, the most successful in one of the criteria may perform poorly in many of the others. The enhancement of all necessary properties is still a challenge for modern materials science and a strong driving force for computational screening modelling efforts worldwide (see Section 4).

Based on principal properties and functionality, protective coatings may be grouped mainly into hard coatings, coatings with enhanced thermal stability, coatings with high oxidation and corrosion resistance and thermal barrier coatings. High hardness is essential for cutting tool coatings. In addition to the usually extended tool lifetime, it helps to achieve a smooth high-quality surface and shape of machined parts. However, hard materials are often brittle and prone to cracking, which is why it is crucial for protective coatings to have both high hardness and toughness. Only a mixture of several parameters leads to high wear performance and the long lifetime of cutting tools under extreme conditions. In the next section, a brief description of the main properties of the hard coatings mainly used to coat WC–Co tools and other tool materials is reported.

#### 2.3.1. Diamond and Diamond-Like Carbon (DLC) Coatings

Diamond is the intrinsically hardest material because of its strong nonpolar covalent C-C bonds (sp^3^ bonding) and short bond length. It belongs to the class of ultrahard materials, with H = 70–100 GPa [99]. Diamond exhibits the highest room temperature thermal conductivity of ∼20 W/cm·K and an extremely low coefficient of thermal expansion of ∼0.8 × 10^−6^ K^−1^ at 300 K. At room temperature, it is inert to attack from acids and alkalis, and it is resistant to thermal shock. The topography of the diamond surface can be very different depending on the extent of polishing and orientation of the crystallographic planes, varying from an extremely smooth surface with a friction coefficient as low as 0.1 in air to a very rough surface with protruding edges [100]. Because of these properties, diamond has tremendous application in the field of tribology, especially as a protective coating for cutting tools. Polycrystalline diamond films are best synthesized by chemical vapour deposition (CVD) techniques (but not limited to this) [101,102]; the most common are hot filament-assisted CVD (HFCVD), direct current plasma-assisted CVD (DC PACVD), microwave plasma CVD (MPCVD) and combustion flame-assisted CVD (CFACVD) [100]. However, most of the metals and ceramics have a much higher thermal expansion coefficient than the very low one for diamond coatings. This often may cause residual stress and the further spallation of the coatings, thus limiting the number of workpiece materials for diamond-coated cutting tools. Tungsten carbide (WC) has the closest coefficient of thermal expansion to that of diamond, which allows diamond film deposition on WC substrates in almost stress-free conditions. Nevertheless, the diffusion of cobalt contained in WC tools promotes higher graphitisation at the diamond–carbide interface, which induces coating delamination during machining. To reduce such effects and improve adhesion, the deposition of an interlayer with a specific composition, dopants, the multilayer architecture of coatings, surface etching and other surface treatment techniques may be applied [103,104,105,106,107,108]. This should also help in the case of the deposition of a diamond coating on surfaces different from WC, with a higher coefficient of thermal expansion.

Diamond-like carbon (DLC), as is clear from the name, exhibits some of the properties intrinsic to diamond. Among them, the most important factors for application in cutting tools are its hardness at a super- and ultrahard level, high wear resistance and low coefficient of friction (∼0.1). Most DLC films are structurally amorphous and can be synthesised by plasma-based PVD and CVD methods. Deposition method and the type of carbon source substantially influence the structural chemistry of the resulting films, which leads to large variations in their properties. Since the 1970s, DLC films have been well developed and discussed [109,110,111]. Structurally, they are made of sp^2^- and sp^3^-bonded carbon atoms and may be classified depending on their structure and hydrogen content as well as the presence of other dopants. DLC coatings with a lower hydrogen percentage (≤40%) and higher proportion of sp^3^ bonding in the structure have a higher hardness. Thereby, hydrogen-free tetrahedral amorphous carbon (ta-C) with the highest sp^3^ content (80%–88%) has an ultrahardness of 80 GPa. DLC films are not only hard but also smooth; furthermore, their tribological properties may be manipulated easily by introducing dopants such as nitrogen, silicon, etc. [112]. They also help to achieve high-quality shapes of the machined surface or cutting edge which are clean, smooth, and without chipping or rounding (see Figure 9). However, it is quite difficult to deposit thick coatings (of 2 µm and more) because of delamination due to the internal stress; in addition, there are limitations on the substrate choice with adherent film. Moreover, such coatings are prone to graphitise at temperatures above 300 °C with a further decrease of their hardness [100]. On the other hand, when the outer layer of DLC is turned to graphite when the tool is in service, both the friction and wear rates should decrease [113].

#### 2.3.2. Transition Metal Compounds

Transition metal nitrides (TMN), carbides (TMC) and borides (TMB) are largely employed as hard protective coatings in the cutting and forming tool industry. They attract interest due to their exceptional properties, such as their high hardness, chemical inertness, electronic properties, high melting point and thermal stability under harsh environments (oxidation, radiation, etc.). Such properties are mainly due to the variety of chemical bonding [114,115,116]. Based on the prevailing bonding type in TM nitrides, carbides and borides, Holleck [117] arranged their properties from low to high levels; some of these are shown in Figure 10. Depending on the priority properties and the suitable deposition technique, the coating which fits an application best can thus be chosen.

The first industrial CVD-coated tool coating on cemented carbide was TiC in 1969, and in 1980 TiN became the first PVD coating. Lower temperatures in the PVD process made the deposition of coatings on steel tools possible [7]. Since the 1970s–80s, and up to the present, TiN, TiC, TiB_2_, CrN and ZrN have been the most frequently used binary coatings.

TiN has been the most widely studied TMN protective coating and has been in wide use since the late 1960s. However, it has some limitations and barely overcomes the modern challenges of thermal stability and oxidation resistance. At temperature above 500 °C, in fact, an oxide layer may be formed on the surface, which develops stress in the coating, which is high enough to damage or destroy the protective layer. 

#### 2.3.3. Multi-Elemental Compounds and High-Entropy Alloy Protective Coatings

To overcome the modern challenges related to performance in extreme conditions, new, more complex compositions of coatings, such as ternary and quaternary compounds, have been proposed, which exhibit some superior and specific properties (e.g., thermal stability and oxidation resistance). An important example of this are Al-containing TMN ternary compounds of the TM_x_Al_1−x_N type, where TM is a transition metal, such as Ti, Cr, Zr, Nb, Hf, Ta, V, etc. The most popular are Ti-Al-N and Cr-Al-N systems, which have remained the “state-of-the art” coatings for a long time and are well-discussed in research and review papers [118,119,120]. Their exceptional hardness and oxidation resistance are mainly caused by the supersaturated solid-solution of hexagonal B4-structured AlN in the cubic B1 structure of TiN, with the resulting large volume mismatch, elastic strain energy and solid solution strengthening [119]. The mechanical and chemical properties of TM_x_Al_1−x_N coatings usually improve with the increase of the Al fraction, but only up to a certain critical Al content (usually around 40%–50%). The formation of Al, Ti or Cr oxides increases the oxidation resistance at elevated temperatures. Moreover, metastable supersaturated films such as TM_x_Al_1−x_N systems exhibit the phenomenon of age-hardening caused by decomposition with annealing temperature or time, leading to an increase in hardness (which is more significant in the Ti-Al-N system than in Cr-Al-N) [119]. This particular feature may have a high potential for the efficient enhancement of tool lifetimes.

Recently, the groups of T. Polcar and A. Cavaleiro performed studies on the addition of Cr to the Ti-Al-N system and its influence on thermal resistance, oxidation stability, tribological and cutting performance by the deposition of quaternary Ti-Al-Cr-N coatings [121,122]. A coating which demonstrates relatively low wear resistance at room temperature (Ti-Al-Cr-N system) in comparison to the other (in this case, the Ti-Al-N system) can exhibit much higher wear performance at elevated temperatures (650 °C) corresponding to realistic conditions for cutting tools’ working temperatures.

Other ternary coatings which combine the superior properties of transition metal carbides and nitrides are transition metal carbonitrides (TMC_x_N_1−x_) such as TiC_x_N_1−x_, whose structure may be described as a TiN matrix with the substitution of N atoms by C atoms, which leads to distortion strengthening and increased resistance to dislocation motion [123]. Depending on the deposition method and deposition conditions, the film may also be composed of a mixture of TiN, TiC and C_3_N_4_ phases [124]. A transition metal nitride contributes to the strengthening and hardening of coatings, while carbon forms a graphite lubricating layer during work and thus substantially reduces the wear rate. 

Apart from Al, other elements can be also added in multielement coatings to confer improved properties. Silicon is very often introduced in nitride coatings to form hard nanocomposite materials, where amorphous Si_3_N_4_ surrounds hard metal nitride grains [125], or a similar multi-layered configuration may be applied [126]. Refractory metals (Nb, Mo, Ta, W, Re) [127] significantly improve the thermal stability of the coatings and allow them to work under extreme conditions and high speed, but most of them are CRM elements (see Figure 2).

More complex compounds such as high-entropy alloy coatings can also be employed as protective coatings. High-entropy alloys (HEAs) are mostly identified as alloys which contain at least five principal elements with a concentration of each between 5 and 35 at.% and the possible inclusion of minor elements to modify the final properties [128,129], which finally depend on the material composition, microstructure, electronic structure and other features in complicated and sensitive ways. HEAs may be deposited as protective coatings as in [130], where Yeh and Lin experimented with the dry cutting of 304 steel with bare TiN, TiAlN, and high-entropy nitrides (HEN) (Al_0_._34_Cr_0.22_Nb_0.11_Si_0.11_Ti_0.22_)_50_N_50_ coated WC-Co inserts and found that only the HEN-coated insert could produce long, curled chips, indicating that the cutting edge of the HEN-coated insert was still very sharp due to its superior oxidation resistance. Because of the great diversity of the possible involved elements in the composition, HEN coatings have high risk of containing CRMs; however, an appropriate screening method could allow the selection of HENs with a reduced or no critical raw material content (see Section 4).

#### 2.3.4. Nanocomposite Super-Hard Coatings

Nanocomposite (nc) coatings are formed by nanometric-sized particles (usually MeN, MeC) embedded in amorphous or crystalline matrices and have attracted considerable interest because of their superior hardness which may allow the challenges of modern cutting tools to be overcome. The reason for their hardness resides in their grain size refinement. These so-called “third-generation ceramic coatings” represent a new class of materials that exhibit not only exceptional mechanical properties, but also excellent electronic, magnetic, and optical characteristics due to their nanoscale phase-separated domains of approximately 5–10 nm. The most recognised explanation for their improved performance is the increase of their grain boundary volume, because grain boundaries impede the movement and activation of dislocations. The so-called Hall–Petch strengthening [89,131] gives rise to high hardness H with a relatively low Young modulus E providing high toughness, enhanced wear resistance, high elastic recovery, resistance against crack formation and crack propagation, high thermal stability (up to 1100 °C) and reduced thermal conductivity [132]. Thus, grain size refinement allows the mechanical and tribological properties of the coatings to be controlled and optimised, and thus helps in increasing the tool’s life. The opposite happens when grain sizes continue to decrease to values smaller than 10 nm, when the maximum hardness is reached. Overgrown volumes of boundaries and small grains cause a loss of material hardness. This phenomenon has been termed as the reverse or inverse Hall–Petch relation/effect. 

There is now growing evidence that a critical H^3^/E^2^ ratio (>0.5 GPa) has to be satisfied for the coating to provide appropriate tribological protection [133] (Figure 11). In this respect, Plasma Enhanced Chemical Vapour Deposition (PECVD) and PVD metal-based nc coatings are very suitable candidates thanks to their optimal tribological properties which may be achieved by opportunely controlling the parameters of the deposition process (deposition time, target power or cathode current, bias voltage applied to the substrate, temperature, pressure, gasses flow, etc.) and the elemental composition of the coatings.

Nanocomposite coatings may provide superior performances than superlattices (multilayers in which a single layer has a thickness not greater than 10 nm), whose properties are strongly dependent on the precise thickness control of the single layers composing the multilayer stack, so that any error mismatch may affect the coating’s performance.

Hard nanocomposite coatings for cutting tools may generally be classified into two groups: (1) nc-Me_1_N/a-Me_2_N (hard phase/hard phase composites), and (2) nc-Me_1_N/Me_3_ (hard phase/soft phase composites), where Me_1_ = Ti, Zr, W, Ta, Cr, Mo, Al, etc. are the elements forming hard nitrides, Me_2_ = Si, B, etc. and Me_3_ = Cu, Ni, Ag, Au, Y, etc. [133]. The most used and interesting coatings for cutting tools are Ti-Al-Si-N with nc-(Ti_x_Al_1−x_)N + a-Si_3_N_4_ phases [134,135,136], Cr-Al-Si-N of nc-(Cr_x_Al_x−1_)N + a-Si_3_N_4_, TiZrSiN of c-(Ti,Zr)N solid solution + a-SiN_x_ [137], nc-TiC + a-C [138,139] Ti-Si-N [140,141], Ti-Si-B-C [142,143,144], Ti-Si-B-C-N [145], AlTiN-Ni [146], ZrN/SiN_x_ [147], nc-W_2_N/a-Si_3_N_4_ [148], (Zr-Ti-Cr-Nb)N [149,150], Mo_2_BC [151], nc-AlN/a-SiO_2_ [152]. In particular, several of these coatings, such as Ti-Al-Si-N, Ti-Si-N and AlTiN-Ni, were deposited and tested on cemented carbide WC–Co-based substrates like in [135,136,140,146]. Nanocomposite hard coatings are well discussed in the most recent and fundamental reviews of J. Musil [133], S. Veprek et al. [153,154], A.D. Pogrebnjak et al. [155,156], C.S. Kumar et al. [157]. 

#### 2.3.5. Multi-Layered and Graded Coatings

Multilayer architecture is one of the most efficient and promising current approaches for the hardness and toughness enhancement of protective coatings. There are several paths to their design: depositing a set of films in a special order according to their functionality (e.g., substrate > adhesion film > superhard film > oxidation resistant film), alternating films with a similar crystal lattice for epitaxial growth, hard crystalline films with thin amorphous layers, alternating TMN, TMC or TMB films, nanocomposites, HEAs coatings, etc. Many research groups around the world have worked on this topic, and many papers have been published that review and evaluate the recent progress in this area [127,133,155,158,159,160]. Among the most recent and interesting multilayer solutions for protective coatings there are TiN/TiAlN [161,162], TiAlN/TaN [163], Ti(Al)N/Cr(Al)N [121], (TiAlSiY)N/MoN [164], CrN/AlSiN [165], AlCrN/TiAlTaN [166], TiSiC/NiC [167], TiN/MoN [168,169], TiN/WN [170], TiN/ZrN [171], Zr/ZrN [172], Ta/TaN [173], CrN/MoN [174,175,176], (TiZrNbHfTa)N/WN [177], and multilayer hard/soft DLC coatings [178]. In particular, the following multi-layered coatings were also deposited on cemented carbide substrates or machining tool inserts, including WC–Co-based inserts, for mechanical tests in laboratories or for industrial tests: TiN/TiAlN [162], TiAlN/TiSiN [179], CrAlSiN/TiVN [180], AlCrN/TiVN [181], TiVN/TiSiN [182], CrN/CrCN [183], AlTiCrSiYN/AlTiCrN [184] TiCrAlN/TiCrAlSiYN [185]. However, K.N. Andersen et al. [162] did not observe differences in the properties of the coatings when using the various substrates (e.g., high speed steel and cemented carbides). Moreover, it should be mentioned that cemented carbide-based substrates and tools usually may be exposed to higher operational temperatures.

At least three significant benefits of multi-layered coatings should be mentioned: the first is the possibility of building two-dimensional nanocomposite multi-layered films with a nanoscale thickness of each individual layer to enhance the mechanical and tribological properties [99]; the second is the adjustment of grain size by changing the bilayer thickness of the coatings, since the grain sizes decrease in thinner layers, as reported in many works; the third and the main intrinsic feature of the multilayer design of coatings is the ability to resist external forces and cracks and to interrupt their propagation toward the substrate on the interlayer interfaces, which prevents the direct impact of destructive factors on the workpiece material of cutting tools (Figure 12 [158]).

A combination of multilayer nc-coatings and cryo-machining was found to improve the cutting performances of the carbide cutting of the Inconel 718 superalloy [186].

Graded coatings are layers whose composition and microstructure are gradually tuned across the thickness from the interface with the substrate towards the outer surface. This kind of coatings, included in the so-called class of FGMs (functionally graded materials), represents an effective way to improve tool performance by enhancing coating/tool properties such as adhesion, toughness, thermal stability, resistance to corrosion, friction, fretting, wear, etc. [187], thus helping to increase the tool life and consequently reducing the consumption of CRMs contained within the tool or the coating itself. In FGMs, a proper gradient in the coating properties (Young’s modulus, thermal expansion coefficient, etc.) can reduce residual stresses due to the lattice mismatch and different thermal characteristics of the substrate and coating, thus decreasing the possibility of coating debonding and delamination. Graded coatings based on nitride, carbide and carbonitride materials such as TiN, TiC, TiCN are commonly exploited for cutting tools based on cemented carbide or other materials containing CRMs such as W, Co and rare earth elements [188,189,190].

Graded coatings are often used as intermediate layers to facilitate the bond between the substrate and a subsequent layer. For example, DLC coatings deposited directly on the surface of a component often result in the generation of high residual stress and consequent poor adhesion, thus hindering the excellent protection properties of DLC described in the previous paragraph. Thus, as shown in [191], an intermediate layer with a composition gradually changing from Ti to TiC (Ti/TiN/TiCN/TiC) can be deposited on a Ti-based alloy to promote the bonding of the final DLC coating. Similarly, a boron-doped graded layer diamond coating, acting as a transition layer between a non-graded boron-doped diamond coating and a nano-crystalline diamond coating, was studied in order to improve the machining performance of tungsten carbide cutting tools [103]. Other complex coating structures which take advantage of the gradient composition can be realized by different combinations of multiple layers, as in [192], where a layered coating was made of a lower fine periodic TiAlYN/CrN multilayer that graded into an upper amorphous TiAlY oxynitride layer in order to obtain enhanced oxidation resistance and reduced friction coefficient in tungsten carbide tools for high-speed cutting applications.

Graded layers can be also obtained by directly inducing modifications into the same surface of the tool; for example, by means of metal ion implantation or gas diffusion/reaction into the surface. In the first case, for instance, Cr ions can be implanted to create a graded layer and metal ion intermixing in the interface region to enhance the adhesion of protective CrN or TiAlN-based coatings on high-speed steels (HSS) [193,194], which are commonly used for cutting tools and contain a significant content of CMRs such as W, V or Co. In the second case, substrate surface modification is obtained by gas diffusion and/or reaction to create a graded layer. Plasma nitriding, carburising or carbonitriding are reported to be effective ways to improve the surface properties of working tool steels and cermets (containing, e.g., W, Co and V) and to enhance the adhesion of nitride or DLC coatings, thanks to the development of a graded diffusion layer and formation of intermediate interface compounds [195,196,197,198]. As an alternative, surface nitriding with a graded composition and microstructure can be achieved on the surface of the material directly during its production process. This technique is particularly useful to enhance the surface properties of WC–Co tools, such as in the case of the so-called functionally graded cemented carbides (FGCCs) or functionally graded hard metals (FGHM), which are a widely studied class of tool materials in which the graded layer is obtained by mixing and pressing powders with a suitable composition, followed by sintering steps under a controlled atmosphere. By properly tuning the different process parameters, the compositional and structural gradient of an FGCC can be tailored (Figure 13) according to the desired properties, improving the tribological properties of the material surface and strengthening the adhesion of additional protective coatings deposited afterwards [199,200,201].

#### 2.3.6. Thermal Barrier Coatings

A high amount of heat is generated in the cutting zone during machining. There are three different zones from where the heat flux comes into the cutting tool: the primary shear zone (plastic deformation and viscous dissipation), the secondary shear zone (frictional and plastic shearing energy), and the frictional rubbing of the cut surface on the tool insert flank. The diffusion of heat into the workpiece or tool body negatively influences the lifetime and work performance. There are many coatings with thermal barrier functionalities when applied to metallic surfaces which lead them to operate at elevated temperatures, but most of them exist for cases with no directly applied high mechanical load, such as gas turbines or aero-engine parts. In case of cutting tools, the influence of the coating on the heat distribution of the working interface is unknown or very poorly studied. It is not clear whether coatings influence the cutting process by an insulation effect (lower heat flux transmitted into the substrate) or rather by a tribological effect (lower level of heat generated by friction) [202]. It is quite difficult to perform in-situ or ex-situ experimental studies of thermal barrier coatings or to find thermal properties data in the literature due to the lack of a standard methodology to quantify these properties in the case of very thin layers. Thus, most of the existing approaches are based on simulation methods, and improvements may be expected by applying the techniques outlined in Section 4.

J. Rech et al. [202] proposed an analytical solution for heat transfer modelling to characterise the influence of a coating on the heat flow entering into the tool substrate. He showed that coatings have no capacity to insulate a substrate in continuous cutting applications, but in applications with a very short tool–chip contact duration, such as high-speed milling, coatings keep a large amount of heat in the interaction zone, which may improve the wear resistance of the tool. Experimental investigations made by the same authors were found in accordance with results from computational studies [203]. In addition, it was shown that the larger the coating thickness, the more it influences heat transfer. Moreover, it has been pointed out that the heat flux transmitted to a substrate is much more influenced by the tribological phenomenon at the cutting interface than by the thermal barrier properties of the coating [202].

M.A. Shalaby et al. [204] reported that the improvement of pure alumina ceramic tool performance in proportion to the cutting speed can be attributed to the thermal barrier properties of the ZrO_2_ tribo-layer induced at a high cutting speed (temperature). In the case of SiAlON ceramic (Si_3_N_4_ + Al_2_O_3_), they pointed out that the high performance was due to the high amount of mullite (Al-Si-O) tribo-film formed on the tool face—a phase that reduces thermal conductivity and serves as a thermal barrier layer. W. Song et al. [205] reported the thermal barrier contribution to the tool wear resistance when coated by Ti-MoS_2_/Zr. Gengler et al. [206] investigated thermal transport for Si-B-C-N ceramic films and concluded that their properties are ideal for thermal barrier applications for high-temperature protective systems in aircrafts, as well as for surfaces of cutting tools and optical devices.

Many protective coatings against wear and corrosion which may be applied for cutting tools are not considered here due to the huge variety of their types and infinite number of specific tasks; however, some general principles and tendencies were discussed, while more information is available in specialised papers and reviews [7,155,207,208,209,210,211]. Although chemical vapour deposition and physical vapour deposition are the two main methods of protective coating fabrication, it is suggested to look deeply into thematic papers and reviews to learn about different techniques [212,213] or find details on deposition methods in research papers concerned with specific materials and structures. 

## 3. New Materials for Tools 

High efficiency often translates into high cutting speeds. Ceramic tool materials are the basis of HSC (high-speed cutting) machining [212]. Ceramic tools show three to ten times more durability than sintered carbide tools, and they can work at least at several times higher cutting speeds. The development of “hard machining” and dry cutting technology is associated with ceramic materials. HSM (high-speed machining) was created as a result of the need to shorten the time taken in manufacturing elements, to eliminate inaccuracies resulting from the use of manual finishing treatments and to minimize manufacturing costs. HSM also enables the fast and efficient processing of hard materials (stainless steels, durable titanium alloys, tool steels), moulds and mould element processing with a high shape and surface precision with low roughness. All these advantages have an impact on simplifying construction work at the design stage. In addition, high-speed machining ensures high removal efficiency, shorter production times, lower cutting forces and reduced deformation of the workpiece as a result of significant heat dissipation through the chips. The above advantages require high rigidity and precision of the machine tool system: the tool, machine work with high spindle speeds and special cutting tools (appropriate tool materials and coatings). Thus, the materials from which the tools are made are one of the most important factors for machining at high speeds and must ensure the tools’ long-term operation. The biggest problem is the negative impact of high cutting speed on the durability of the cutting blade. The most important features of tool materials for cutting blades are high hardness and abrasion resistance so that the tool does not require frequent regeneration. The necessary property of the cutting material is high resistance to dynamic loads and brittle fracture. The latter will protect the cutting blade against chipping (arising, for example, from the heterogeneity of the material’s properties or the insufficient stability of the machine spindle). Due to the heating of the tool during operation, the material should retain its properties in a wide range of temperatures. For this reason, a favourable feature of the material is a high thermal conductivity coefficient, since in all cases of machining, it is possible to use cooling liquids. Cooling liquids are troublesome because they contain particles of often harmful elements that come from the workpiece. Sometimes, in the case of organic liquids (currently displaced by synthetics), these liquids can be bacterially contaminated. Nevertheless, an important feature of the tool material is the chemical resistance, excluding the possibility of chemical reaction with the material being processed, and thus its corrosive destruction. In addition to the features mentioned above, materials intended for cutting blades should be machined in an effective manner, ensuring proper shape and dimensions, especially for tool blades. All ceramic cutting tools have excellent wear resistance at high cutting speeds. There are a range of ceramic grades available for a variety of applications.

### 3.1. Basic Groups of Tool Materials Intended for Cutting

Ceramic tool materials have a stable position in terms of their range of application and range of species. It is estimated that they currently constitute 9% of all tool materials [9,213]. They are offered in the form of the mechanically fixed indexable inserts used by all major tool companies, regardless of several companies which specialise exclusively in tool ceramics (Figure 14).

The following groups of ceramic tool materials are distinguished here:
Oxide ceramics are based on aluminium oxide (Al_2_O_3_). This material is chemically very stable, but lacks thermal shock resistance. Due to the low price and resistance to abrasion at high temperatures, it is used in the medium-fine machining of cast irons with a Brinell Hardness below HB235, carbon steels with a Rockwell Hardness the C scale lower than HRC38, as well as alloy steels. Most often, pure Al_2_O_3_ is used for machining parts made of grey cast iron for the automotive industry.Mixed ceramics Al_2_O_3_ with the addition of ZrO_2_, TiC, TiN, or TaC, NbC, Mo_2_C, Cr_3_C_2_: the most popular method of strengthening Al_2_O_3_ is the introduction of ZrO_2_. The polymorphic ZrO_2_ transformation occurs at 1150 °C and results in an increase in the volume of the zirconium-containing phase. The change in the volume of the ZrO_2_ phase generates in the Al_2_O_3_ matrix the stresses which are able to absorb the energy of the cracks. Mixed ceramics are particle-reinforced through the addition of cubic carbides or carbonitrides. These additives improve the toughness and thermal conductivity of the material. The materials are used to anneal iron alloys with a hardness of 55–65 HRC, including cast irons, brittle materials such as composites on metal matrix-reinforced ceramics or intermetallic materials and high-density alloys based on tungsten. These materials can also be used in continuous and intermittent processing and milling and turning as well as roughing and fine machining conditions.Whisker-reinforced ceramics use silicon carbide whiskers (SiCw or Si_3_N_4_w) or single-leaf monocrystals, most commonly SiC. The critical stress intensity factor K_Ic_ for such materials is from 8 to 10 MPa m^1/2^, and the bending strength is in the range of 600–900 MPa. These materials are used for machining with low cutting speeds of nickel alloys, hardened steels, non-metallic fragile materials and high hardness cast irons [214].Nitride ceramics: Si_3_N_4_ with additives to facilitate sintering, and SiAlON. Si_3_N_4_ elongated crystals form a self-reinforced material with high toughness. Silicon nitride grades are successful in grey cast iron machining, but their lack of chemical stability limits their use in other workpiece materials. Materials based on silicon nitride have a toughness similar to hard metals and the temperature resistance characteristic of oxides. This extends the scope of their applications and allows them to be used, for example, for the roughing and semi-finishing of cast iron castings with turning and milling as well as in the machining of special alloys with high nickel content. SiAlON grades combine the strength of a self-reinforced silicon nitride network with enhanced chemical stability. They are ideal for machining heat-resistant super alloys (HRSA).Superhard materials: diamond and regular boron nitride. These materials are designed for machining difficult-to-cut materials. The most commonly used are polycrystalline sintered materials. Diamond is used for non-ferrous metals, and regular boron nitride is used for hardened steels.

Silicon metal and borate belong to the critical raw materials list, as natural graphite. However, various carbon sources can be used in the production of diamonds. Bearing in mind the limitation of the use of critical materials in materials intended for machining, the development of tools from Al_2_O_3_ and diamonds has been taken into account later in this chapter.

Ceramic materials have some disadvantages: they are not easily amenable to machining, and it is difficult to give them a complicated shape. Therefore, cutting inserts made of these materials usually have the shape of a circle, triangle, rhombus or square (Figure 14). Thanks to the advancement of materials engineering, there are many new materials for which machining technology needs to be developed. Most companies offering cutting inserts provide catalogue information in which the following materials are distinguished: unalloyed steels, low-alloy steels, high-alloy steels, stainless steels, tempered steels, cast irons, titanium alloys, nickel and cobalt, aluminium alloys, wood, and sometimes polymers and graphite.

### 3.2. New Considerations for Cutting Tool Materials

The development of civilization imposes the necessity of new solutions. New factors have to be taken into consideration which have not been considered properly to date. The basic factors are, of course, health and the human environment and the availability of resources; thus, the use of CRMs should be avoided as much as possible. Certain types of materials have been and are being used despite their harmful effects. The reason for this is the lack of substitutes. This is the driving force behind undertaking research work into new and better solutions. When searching for new substitutes, toxic, allergenic and carcinogenic materials or materials that can work only in the harmful environment of lubricating and cooling liquids should be avoided. Limiting the use of harmful cooling liquids is possible because of the higher thermal resistance of new cutting materials. Lower friction and wear materials reduce energy consumption and enhance the lifetime of used tools, thus saving materials and allowing a reduction of the used lubricants, or even enabling the use of dry machining. Coolant and lubricant costs account for 16% of the total machining cost, and tool costs only account for 4%. Therefore, the avoidance of cooling will have a high benefit and impact by the reduction of lubricant costs due to low-wear and low-friction ceramics. Human contact with harmful compounds may occur at various stages; e.g., material preparation or treatment with a blade made of this material (in the form of an inhaled aerosol), or as a result of the utilization of the tool material. 

Ceramic tool materials usually have the ability to be machined at much higher cutting speeds compared to cemented carbide materials; e.g., the cutting speed should be properly selected to produce adequate heat in the cutting zone for chip plasticization, but it cannot be too high, or the chemical or phase composition or selected properties of the ceramics used will be changed. Higher feeds and cutting depths require a reduction in the cutting speed. 

Ceramic materials can substitute cemented carbide tools. Currently, these cemented carbides account for about 50% of total tool production. Cemented carbides are popular as there is a large variety of commercial cutting tools made of them at a relatively low cost. In addition, they are easy to shape, and their operation parameters are consistent. However, without coating, they operate at lower machining speeds than materials with an Al_2_O_3_ matrix. When coated, the edge sharpness of the cutting tool may become an issue. Furthermore, they have low resistance to oxidation, they require intensive cooling, their cutting speed is lower than that of the ceramic materials, and their prices change frequently depending on the supply situation of their raw materials.

Today, there are several ways to develop new tool materials. One is the possibility of using new sintering techniques—for example, SPS/FAST (spark plasma sintering/field-assisted sintering technology) or ultra-high-pressure methods—guaranteeing the high density of sintered materials [156,157,215]. These methods allow us to obtain new materials which were previously not used as tool materials; this group includes carbides and high-melting borides or nanoceramics. The conventional sintering of ceramic requires high temperatures and long times for densification; therefore, the lowest grain size achievable by this technique remains of about 0.5 mm. There is the need to investigate alternative sintering methods that enhance mass transport and make it possible to lower both the temperature and the time of consolidation and, thus, to control grain growth. For these techniques, the sintering of extremely refractory materials is possible, or, as an alternative, the lowering of the temperature of consolidation. SPS employs a pulsed DC current to activate and improve the sintering kinetics. Three mechanisms may contribute to field-assisted sintering: the activation of powder particles by a pulsed current, which leads to the cleaning and surface activation of powders, resistance sintering and pressure application. Thus, the densification of nanopowders occurs at temperatures significantly below those of larger-grained powders by up to several hundreds of degrees. Consequently, small final grain sizes may result, and sintering aids and undesirable phase transformation may be avoided [216]. Materials characterised by a nanometric particle size have better resistance to chipping and better mechanical properties. There are several publications documenting the great success in the field of synthesis of diamond and diamond composite materials sintering using SPS technology, which is cheaper than the high-pressure–high-temperature methods [217,218,219,220]. For example, the combination of an ultrafast heating rate of about 2000 °C/min and a short holding time for SPS was successful in fabricating diamond/cemented carbides. For most experiments, the size of the diamond was over 40 μm, and the diamond content was limited to up to 30 vol.%. Several experiments show that it is possible to prevent the diamond graphitization and improve its bonding strength with the matrix. The SPS/FAST method guarantees a reduction of the manufacturing cost, in comparison to the high-pressure–high-temperature sintering used for polycrystalline cBN sintering. The laser beam sintering of ceramic bodies is also a promising technique for new ceramic tools [221].

### 3.3. New Solutions for Tool Materials

The commercial ceramic tool materials presented above have many drawbacks and are imperfect. The fracture toughness of the materials is low because the dislocation movement is extremely limited by their ionic and/or covalent bonds. The brittleness and poor damage tolerance have so far limited their application as advanced engineering materials, especially in cutting applications. The problem is that the low thermal shock resistance of these materials implies using a huge amount of cooling liquid during the machining process. Good cutting properties depend on the thermal resistance of the material or the use of lubricant–coolant process media. Innovative solutions for cutting tool ceramics will result in the development of ceramics exhibiting higher hardness, fracture toughness, heat and wear resistance in comparison with previous solutions. New materials will have high thermal stability, high corrosion and wear resistance, high fracture toughness and will be obtained with a view to fully replacing the presently available commercial WC–Co hard materials. New Al_2_O_3_ matrix composites, with the inclusion of various novel reinforcing phases—for example, cBN (cubic boron nitride) [222] or Al_2_O_3_ fibres coated with graphene—will substitute hard metals (sintered carbides) [223,224]. It is evident from this discussion that the computational techniques outlined in Section 4 could play a major role in designing tool materials with improved mechanical properties.

A distinct change in properties is obtained by using metals, especially in the form of nanometric additives [201]. The simplest way of introducing metallic additives to Al_2_O_3_ is the metallization of powders by mechanical means through reduction or electrochemically. Cutting fluids are widely used in machining processes, especially for ceramic cutting tools. The main roles of cutting fluid are cooling, reducing friction, removing metal particles, and protecting the workpiece, the tool and the machine tool from corrosion. A new idea is to obtain self-lubricating ceramic cutting tool materials with the addition of solid lubricants. Solid sliding agents should be characterised by a low friction coefficient and resistance to oxidation at temperatures exceeding 800 °C [225,226,227,228]. 

One of the interesting methods to obtain self-lubricating ceramic cutting tool materials is the addition of metal-coated solid lubricant powders. Nickel coated CaF_2_ composite powders with a core–shell structure were produced by the electroless plating technique [229].

For cutting tool edge applications, diamond polycrystals have been used since the early 1970s. Katzman and Libby have reported the liquid phase sintering of a diamond–cobalt system [227]. Hibbs and Wentorf have developed a method of cobalt infiltration into diamond layers under high-pressure conditions [230]. Diamond powders were sintered and bonded to a WC–Co substrate at the same time by the infiltration of Co from the substrate during the sintering process. The carbide substrate is very useful for tool producers because of the possibility of brazing to the tool body. The sufficiently high wettability of diamond materials by molten metal fillers is the principal requirement for successful brazing. The most popular commercial PCDs are two-layer materials with a cobalt phase. Cobalt provides a good wetting of diamond crystallites; this property allows the production of polycrystals characterized by a low amount (below 10 wt.%) of the bonding phase, resulting in high hardness. Cobalt-containing PCDs are chemically stable only up to 700 °C, while their working temperatures may rise even higher. The presence of a cobalt phase in the diamond layers has a strong influence on the decrease of thermal resistance. The thermal stability of a PCD material can be defined as the resistance to graphitization in an inert atmosphere at elevated temperatures. The fields of interest are diamond polycrystals with higher thermal resistance. The thermal resistance of diamond composites depends on the oxidation process more than on the graphitization process. The evolution of CO gas during diamond composite oxidation destroys the integrity of the composite microstructure. One of the possibilities to increase the thermal resistance of PCD materials is to reduce the cobalt bonding phase content by etching a metallic additive from the PCD layer after sintering [231]. A second method is the preparation of the materials with a non-metallic bonding phase, without the negative effects caused by the diamond–graphite solvent/catalyst on diamond graphitization [232,233]. It was confirmed that the introduction of ceramic phases, with high-temperature resistance, improved the resistance of polycrystalline diamond to oxidation and graphitization at high temperatures [234] (Figure 15).

The most popular method of obtaining bulk diamond compacts is the high-pressure–high-temperature sintering process. The development of multianvil apparatuses allows the use of pressures above 10 GPa even in commercial PCDs. Thus, it was possible to obtain a single-phase polycrystalline diamond by using direct conversion. This material is characterized by excellent temperature resistance and resistance to abrasion [235,236,237]. Other possible substitutes for cemented carbide tools could be materials based on intermetallics. Due to their interesting combination of properties, such as their relatively high fracture toughness and hardness, the anomalous temperature dependence of the yield strength [238] and oxidation resistance [239,240], aluminides of iron, nickel or titanium are the most promising and the most frequently mentioned candidates for the matrix of a composite tool material. Since aluminides form a natural oxide layer of aluminium oxide when exposed to corrosive or high-temperature environments, aluminium oxide seems to be the most suitable reinforcement for aluminides. The wear resistance of NiAl-Al_2_O_3_ composites prepared by self-propagating high-temperature synthesis (SHS) was studied in 2011 [241]. It was found that the composite has very good wear resistance in the case of the addition of 10 wt.% of Al_2_O_3_ short fibres. The wear behaviour was found to be comparable with AISI D2 tool steel, but it did not reach the performance of cemented carbides. In order to improve the properties, the more complex materials of NiTi-TiC-Al_2_O_3_ [242] and NiAl-AlN-Al_2_O_3_ were tested [243]. However, these materials were developed mostly as high-temperature alloys, and hence the wear behaviour is not known yet. On the other hand, the use of carbide reinforcement in nickel or iron aluminide provided highly interesting results. In the case of FeAl–WC composite tools, their successful application in the dry machining of copper bars has been reported [244]. If boron is added to a FeAl–WC composite, then it reaches a higher fracture toughness, higher hardness and wear resistance than WC–Co [245]. Composite materials of Ni_3_Al–WC also achieve a comparable wear rate and fracture toughness to WC–Co [246]. This means that the cobalt in cemented carbides could be well substituted by iron or nickel aluminide. If we would also like to substitute the carbide phase, titanium carbide seems to be the best solution. FeAl-TiC has a fracture toughness approaching that of WC–Co (10.8 MPa·m^1/2^) and a comparable hardness to the cemented carbide [247]. The wear resistance data for this material are not available yet, but it can be expected that they will be comparable to WC–Co due to the similarity of the other properties. Very good wear resistance, comparable to cemented carbides, was achieved in the case of Ni_3_Al-TiC [248]. As presented above, aluminides could potentially substitute cobalt binders in cemented carbide and provide a similar wear behaviour as well as fracture toughness. However, there is one intermetallic phase which even has a much higher fracture toughness than aluminides: the NiTi phase. It is reported that its fracture toughness is around 40 MPa m^1/2^ [249] in combination with a hardness of 303–362 HV [250]. The nitinol alloy, based on the NiTi phase, is used worldwide in medicine for stents or braces and in electronics for thermal switches due to its shape memory and superelasticity effects. It has been proved that it could be used also as a matrix for a composite, providing superior wear resistance and toughness, reaching the parameters of cemented carbide [251]. Since there are intermetallics with a high variety of properties, there are also ideas to develop a tool material based on the combination of softer intermetallics and harder ones, which act as the matrix and reinforcement, respectively. This approach was tested on the examples of TiAl-Ti_5_Si_3_ [252] and Ni_3_Al-Ni_3_V [253]. While the TiAl-Ti_5_Si_3_ exhibits very high hardness and wear resistance, but low fracture toughness [254], the Ni_3_Al-Ni_3_V has already been applied successfully as the tool for the friction stir welding of stainless steel and titanium [253]. High-entropy alloys, as novel candidate materials for cutting tools, are receiving increasing attention. Recent studies have investigated the substitution of Co binders in cemented carbide cutting tools by HEAs [255]. The search for Co substitutes is an active research field, which is demonstrated by the numerous papers recently reported [256,257,258]. The use of HEA binders with a reduced or no critical raw material content together with the opportune choice of the carbide phase eventually made by HEAs with reduced CRMs content may provide an alternative to WC–Co cemented carbides.

## 4. Modelling and Simulation 

As stated in Section 1, the partial or total substitution of CRMs is one of the several approaches implemented within the RMI to try to deal with the scarcity and external dependence of these highly strategic materials. Even if the attainment of an evetual complete substitution is just an impractical dream in the most difficult cases and in the best of scenarios, it demands tremendous efforts from researchers and technologists [259,260]—the term “substitutability” has even been coined at Yale (Figure 16)—however, it is nevertheless worth trying to analyse in depth how to progress to a gradual and relative reduction of the most critical components of a given piece of machinery. In that regard, several computational modelling approaches have been considered in recent years to be the most appropriate by the experts in the field, all of which come under the umbrella of the multiscale, multiphysics and multidisciplinary modelling paradigm. Generally speaking, they may be divided into three broad categories: Data mining, computational screening and high-throughput machine learning-related tools.

### 4.1. Data Mining

Given the immense amount of computational results that have been produced in the last 30 or more years, and especially beginning with the advent of the new century, several research groups have decided to join efforts by creating consortia responsible for developing and maintaining scientific research data repositories capable of storing the essential input/output results uploaded by materials scientists themselves, concurrently facilitating the cost-free retrieval of large amounts of information by interested researchers worldwide. The diversity of formats and quality of the available files and folders have led to an intense homogenisation strategy to include information which is as clear as possible on provenance and tracking. Due to the obvious links to original US (Materials Genome Initiative [261]), European (Psi-k [262], CoEs (Centres of Excellence in HPC applications [263], EXDCI (European Extreme Data & Computing Initiative, [264]) and Japanese (National Institute for Materials Science, NIMS [265]) undertakings, the databases and repositories are currently somewhat biased to electronic structure simulation data (NoMAD [266] and FAIR-DI [267], AiiDA [268], MaterialsCloud [269], CMR [270], MPDS [271], OMD [272], ESL [273], ESP [274], Materials Project [275] (Figure 17), AFLOW [276], OQMD [277], MatNavi [278]). However, several other initiatives are acquiring momentum as data mining techniques have become state-of-the-art frameworks in materials’ design investigations; among them are several extensions of the above, as well as products—both open-source and proprietary—dealing with upper rungs of the multiscale ladder (CALPHAD and related [279,280,281,282], MMM@HPC [283], EUDAT [284], Kaye and Laby [285], MEDEA [286]), and more general (NanoHUB [287], pymatgen [288]) or more specific properties or systems (Imeall [289], MPInterfaces [290], pylada [291]), to give just a few examples of the impressively growing catalogue of the field. Obviously, extracting relevant information from these computations could be extremely hard without the highly computerized retrieval (data mining) software incorporated as another tool for the repositories; this benefits interested researchers, who may thus save huge amounts of CPU resources and time when discarding or focusing on the partial substitution of a certain element or material for another less critical one with similar or even improved specific properties tailored to a particular application.

Figure 18 shows an example of data retrieval from two heavily used and pioneering repositories after searching for W–C–Co. Concerning computational screening (and its parallel approach, more experimentally biased, combinatorial screening, which has been used for a long time at least via a trial-and-error methodology, as this is essentially the procedure undertaken by humanity to create the very first alloys, namely bronze, CuSn, and brass, CuZn), the rapid advancement of computer capabilities and software has made possible the assessment of the performance of even slightly changed alloy compositions in the search for improved properties. For more than two decades, binary and ternary alloys using practically every element in the periodic table have been tracked and their phase diagrams described with unprecedented accuracy, both experimentally and computationally ([293,294,295]. The emergence of multi-component alloys (MCA) or (single-phase) high-entropy alloys (SP-HEA) in the present decade has boosted an immense and renewed interest in the stability, structural, electronic, elastic, magnetic, and thermodynamic properties of these combinations of several elements, none of which, in contrast to preceding alloying methodologies, can be said to be predominant [296,297,298,299,300]. 

### 4.2. Computational Screening

Genetic (USPEX [301], Figure 19a, GASP [302]), particle swarm optimization (CALYPSO [303]), cluster expansion (ATAT [304], UNCLE-MEDEA [305]), neural networks (RuNNer [306]), and random algorithms (AIRSS [307]), among others, have awakened the creativity of materials scientists, metallurgists, physicists and chemists alike in an unprecedented effort focused on the design of new materials containing smaller quantities of critical elements without compromising their ranges of applicability. As an example of the approach that may be pursued, the new superhard material WB_5_, representing a possible alternative to tungsten carbide, although it still contains critical W, provides excellent performance even at high temperatures, as predicted by the global optimization code USPEX [308], as seen in Figure 19b.

### 4.3. Machine Learning

Machine learning is a cross-cutting methodology that has recently been developed in connection with the huge increase of computational power and advanced numerical and analytical techniques. In intimate connection with artificial intelligence concepts, appropriately configured computational programming and access to high-throughput resources allows a machine to extract patterns and learn from pre-existing data bases much faster and more accurately than ever before, iterating the processes until fully satisfying relationships and results have been obtained and, in doing so, reducing human intervention to a minimum. Interatomic potential fitting for ulterior MD modelling (Atomistica [310], Atomicrex [311], Potfit [312], OpenKIM [313]), hybrid DFT(Density Functional Theory)-MD simulations (Gaussian approximation potential (GAP) [314], SNAP [315]) or DFT-KMC [316,317] and finite element [318,319,320] multiscale modelling approaches are paradigmatic examples of advanced materials simulation methodologies that make use of machine learning-related techniques. The scaling up from ab-initio and atomistic time and lengths to mesoscopic and macroscopic scales benefits enormously from these kinds of procedures, as stated in several different recent reviews (Figure 20).

### 4.4. Summary

In summary, the simultaneous application of the three aforementioned categories to a problem as intrinsically multiscale and multidisciplinary, as is the topic of the present review, provides innumerable advantages: one may envisage researchers willing to partially substitute a critical element (tungsten, cobalt, etc.) present in advanced cutting tool machinery. First, they explore the available similar compositions via computational materials repositories in a data mining process that builds on previous simulation results; secondly, once possible improvements in pre-existing alloy compositions have been identified, they wisely use computational screening techniques that will significantly reduce the number of possible variants, while crucially contributing to the identification of stable and metastable phases that deserve exploration while excellent working performance is not jeopardised; and thirdly, in the process, they build up a complete structural, electronic, elastic, magnetic database that in turn provides excellent quality input results to feed simulation techniques focused on higher lengths and times in an effort to advance to engineering scales and inform experimenters regarding the most industrially promising alloys with a lower critical materials content that could be amenable to sample synthesis in the laboratory and eventually reach the market.

## 5. Recycling 

The recycling of hard metals and coating materials made of hard metals is an important element in the circular economy and environmental [321,322] and worker [323] safety. In terms of the life cycle of cutting tools, recyclability is also derived from a cost of ownership evaluation based on life cycle analysis tools [324]. 

Ishida et al. [325] have stated that if the volume of industrial production continues unaltered, global tungsten springs worldwide would be removed in about 40 years. Therefore, the recycling of tungsten from waste is becoming increasingly important. According to 2013 data from the International Tungsten Industry Association (ITIA), the recycling rate was 50% in Europe and the United States, in contrast to 30% in Japan. To improve the tungsten recycling rate in Japan, an efficient method could be to use scrap cemented carbide because the tungsten content in cemented carbide is more than 80%, which is higher than from other uses [326]. In fact, the Sumitomo Electric group focused on recycling tungsten from used carbide tools and succeeded in developing a technology and making a business. Various recycling processes for WC–Co cermets from cutting tools, such as chemical modification, thermal modification, the cold-stream method and the electrochemical method, have been investigated, and some of them are actually employed in industry. Presently, industrial recycling technology is committed to recycle tungsten carbide by two general methods. The first is the direct method, in which the binding metal is separated from the cemented carbide preserving the same composition; the second is an indirect method in which the dissolution of the binding metal takes place. In direct recycling techniques, such as the zinc process or the cold flow process, consumption is reduced and process costs are low, and the carbide scrap recycling rate is high. Instead, indirect recycling processes use acids and electrochemistry to dissolve the binder phase in cemented carbide waste. With this method, the energy consumption is high, process costs are high and the carbide scrap recycling rate is low. One of the main sources of the world supply of tungsten comes from the recycling of tungsten carbides, and it is believed that 30% of this comes from the recycling of tungsten carbide waste, mainly from tools such as cutters, turning, grinding and energy waste [325]. The used carbide inserts are also converted into powder by a process called the zinc process. In total, 60% of recycled tungsten comes from the recycling of high-speed steels. It is believed that tungsten recycling is much cheaper and more environmentally friendly than scrap disposal [326]. However, the industrial world continually requires the improvement of conventional methods and development of ever cheaper and more efficient recycling procedures. E. Altuncu et al. [327] investigated the applicability of the zinc-melt method (ZMM) for recycling WC–Co as a powder from cutting-tool scraps. It was proven that ZMM is an applicable technique for recovering the WC powder from the cutting tools. WC–Co powders are recovered and then spray-dried, sintered and obtained as a feedstock material for the thermal-spray coating processes. The zinc melt method can be used for the recycling of WC products, and the recycled powder with the addition of fresh powder can be used in the manufacturing of new samples, which shows that the best results are obtained with 70% of recycled WC powders. When identifying the need for customized tooling solutions using hard metal alloys, additive manufacturing can be chosen as a new production process thanks to its superior material and process flexibility. 

Lee et al. developed a mechanochemical approach for WC recycling which involves a reaction with NaOH in a grinder [328]]. 

V.V. Popov et al. [329] studied the effect of powder recycling on Ti-6Al-4 V additive manufacturing. It was shown that as-printed samples produced from the recycled powder have dramatically decreased fatigue properties and a shortened lifetime. However, hot isostatic pressing (HIP) was suggested as a method to improve the quality of the sample and to achieve microstructural and mechanical properties very close to those of samples made of new powder.

Joost et al. suggested a recycling technique [330] which comprises a carbothermal process of simultaneous WC reactive sintering in a presence of graphite. 

The electrochemical processing of WC–Co materials suggested by Malyshev et al. [331] offers good perspectives for the separation of the WC phase from hard metals and ore concentrates. 

Acidic leaching is an up-to-date hydro-metallurgical approach for metal recycling including hard metals processing; however, more research is needed to increase the efficiency of reactions [332].

A microwave approach for recycling hard metal tools produced by chemical vapour deposition technique was proposed by Liu et al. [333].

A. C. Van Staden et al. [334] studied the SLM process using a cemented tungsten carbide powder for tools. The laser power, scan velocity, and hatch spacing of the SLM process were varied, and single powder layers were sintered accordingly. This was done to determine the influence of these parameter combinations on the melting behaviour of the material during sintering. It was found that a combination of high laser power, high hatch spacing, and low scan speed yielded the best results. It is hoped that tool manufacturing can soon be developed using additive techniques starting from recycled tungsten carbide powders.

## 6. Conclusions

The dependency of the EU on imports of CRMs—in particular, in the mechanical manufacturing industry—has raised a variety of concerns about the availability of raw materials and their rational use in tools manufacturing. 

Due to the high import amount of CRMs needed for the manufacturing industry (such as tungsten, chromium, and niobium, to name just a few), it is vital to develop alternative solutions to face the high supply risk posed by the countries that are monopolistic suppliers of these CRMs.

Thus, several solutions must be pursued, such as improvements in strategies for longer-lasting tools (e.g., processing strategies and protective coatings), the development of alternative materials constituting the tools, the improved use of cutting-edge simulations and advanced use of recycling methods, as well as finding suitable candidates to partially or totally substitute the CRMs and increase their recycling.

European initiatives, such as the COST Action CA15102 “Solutions for Critical Raw Materials Under Extreme conditions”, aim at developing strategies for the effective use of raw materials, including the reuse and recycling of end-of-life products containing CRMs.

The present review describes several strategies to face the CRMs issue in machining tools:The development of new cemented carbides based on environmentally harmless binders is a current research area. The substitution of cobalt in the cemented carbides is one of the research trends in the area of the environmental sustainability of industrial production and recycling processes.Different alternatives to the typical tungsten carbide material have been examined as constituents for the machining tools, such as ceramic materials, diamond-based systems, intermetallic systems and high entropy alloys, together with their related effective production techniques.We presented advanced machining techniques such as methods aided by laser, cryogenic temperatures, vibrations and surface defects with the aim of extending the tool life-span and thus reducing the amounts of CRMs used in the tools.We reviewed protecting coatings, which enable an increase in tool lifetimes under different machining situations.Additive manufacturing technologies along with the extensive use of advanced cost-effective fast-track computational methodologies facilitate the development of new materials by opening new ways of designing tools without or with only the partial use of CRMs with efficient strategies for the easy recycling of raw materials.Novel methodologies of tools manufacturing with geometric structures and challenging sub-structures and the potential to produce under-functionally different structures and optional materials with graded properties (e.g., the 4D-printing approach) provide a reasonable approach for decreasing the amounts of CRMs in tools.

In summary, every approach discussed in this review should result in the better quality of the machined parts, improve the machining performance and reduce the use of CRMs. However, in view of the conscious use of raw materials, it is clear that the different strategies should be combined together in a synergistic manner with a new way of thinking regarding the final products through the eco-friendly design of materials, tools and production methods to foster the integration of good practices and habits in the circular economy.

## Figures and Tables

**Figure 1 materials-13-01377-f001:**
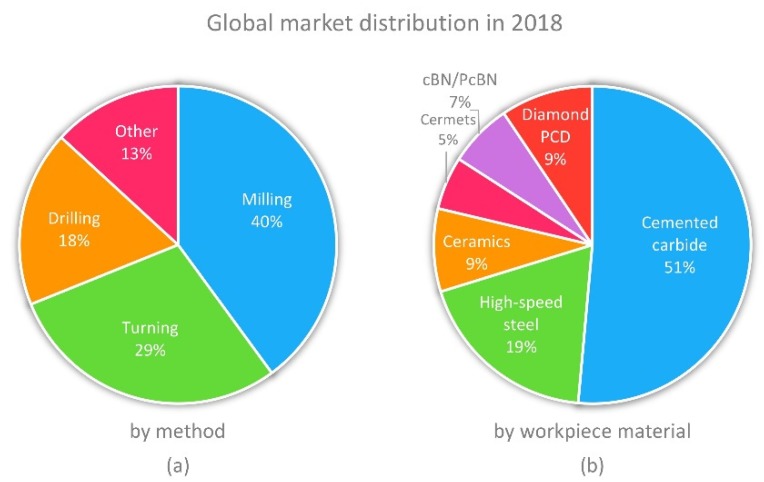
Cutting tool global market distribution by cutting technology (**a**) and by workpiece material (**b**) as presented by Dedalus Consulting. Data taken from [7].

**Figure 2 materials-13-01377-f002:**
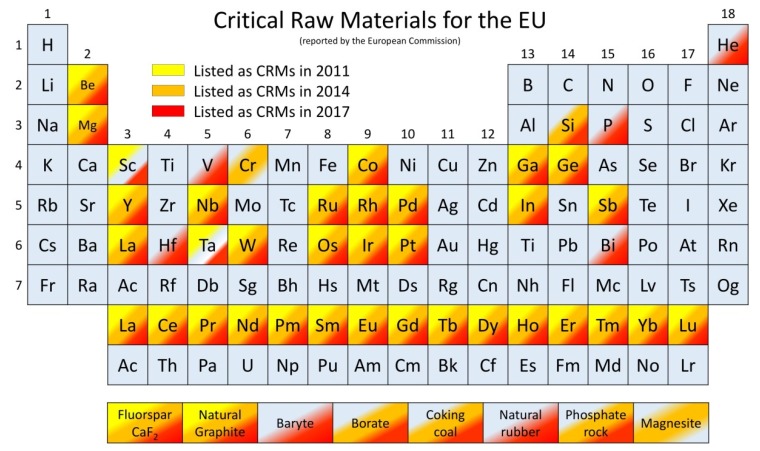
Critical raw materials list for 2011–2017 overlaid on the periodic table of the elements [5,6].

**Figure 3 materials-13-01377-f003:**
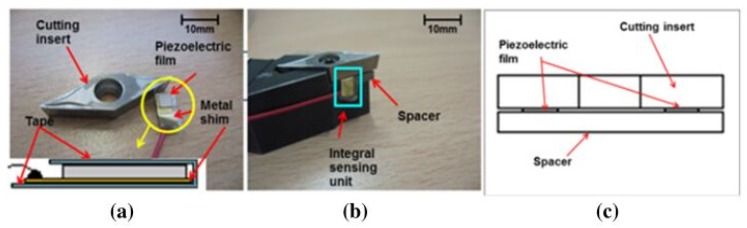
(**a**) Composition of the smart cutting tool, (**b**) assembly of the tool, (**c**) cross-section view of the tool, as taken from [10].

**Figure 4 materials-13-01377-f004:**
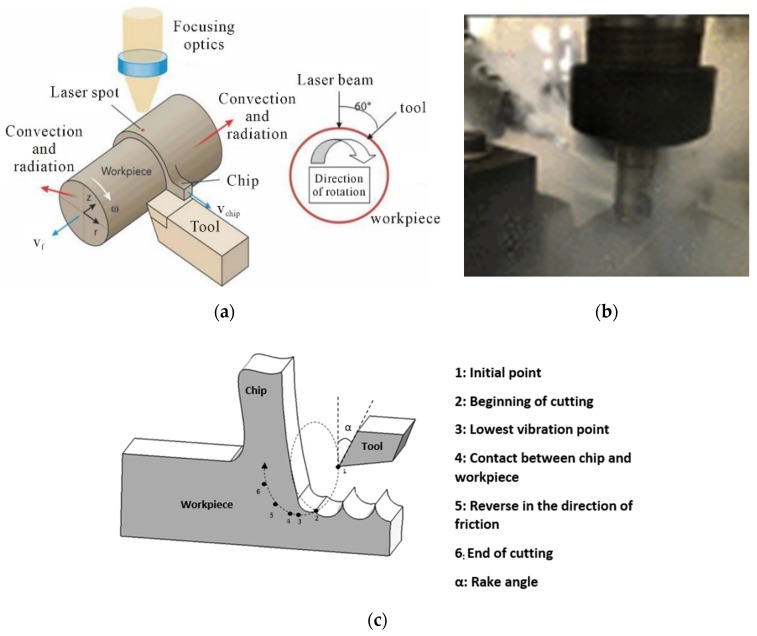
Important advancements made in machining technology. (**a**) Thermal-assisted machining [46]; (**b**) cryogenic machining [47]; (**c**) schematic diagram to illustrate the mechanism of vibration-assisted machining [48].

**Figure 5 materials-13-01377-f005:**
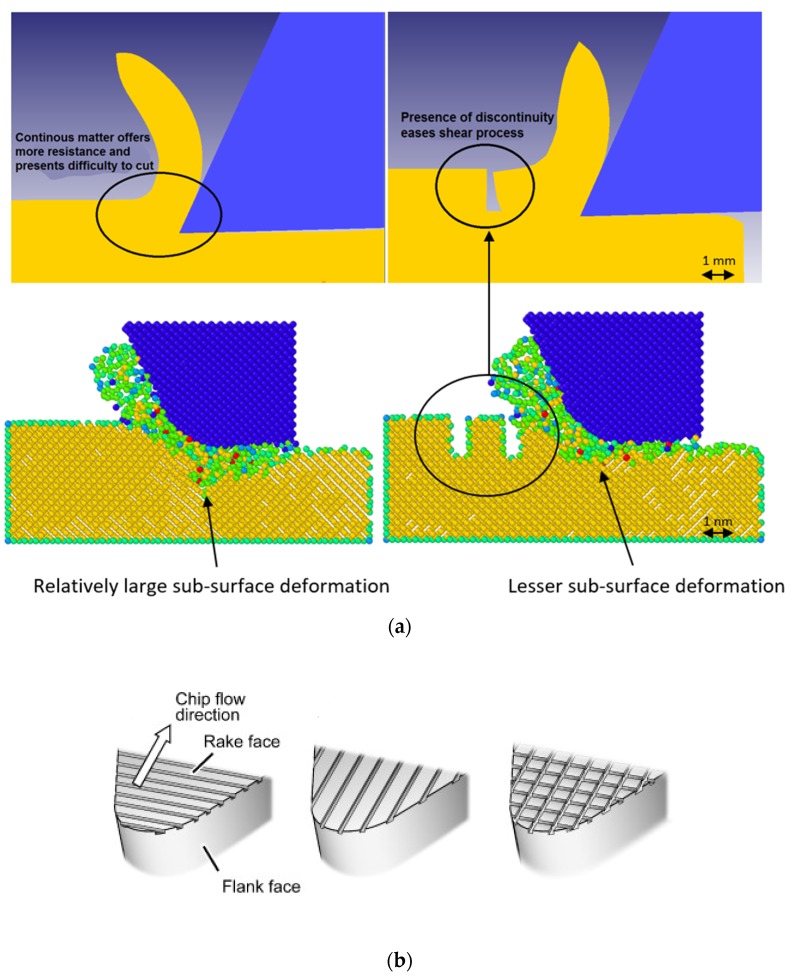
(**a**): Schematic diagram indicating the differences between the mode of deformation during conventional machining and surface defect machining (SDM) observed through an FEA (Finite Element Analysis) simulation of hard steel and MD (Molecular Dinamics) simulation of silicon carbide, respectively; (**b**) effect of providing nanogrooves on the tool [78]. (**a**) Surface defect machining. (**b**) Providing nanogrooves on the tool.

**Figure 6 materials-13-01377-f006:**
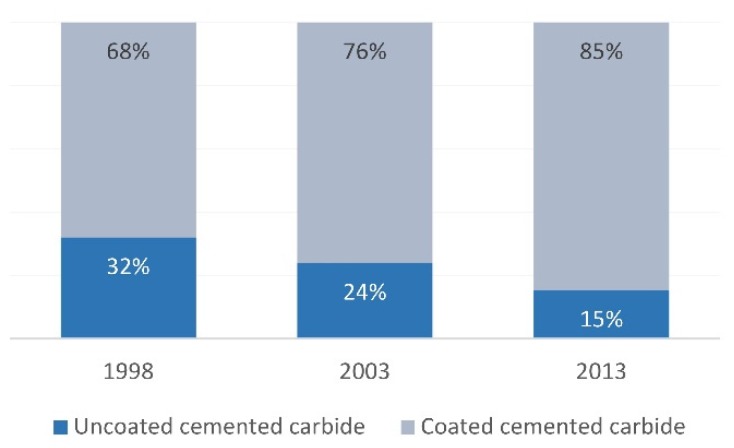
World market of cemented carbide cutting tools. Data from Dedalus Consulting, taken from [7].

**Figure 7 materials-13-01377-f007:**
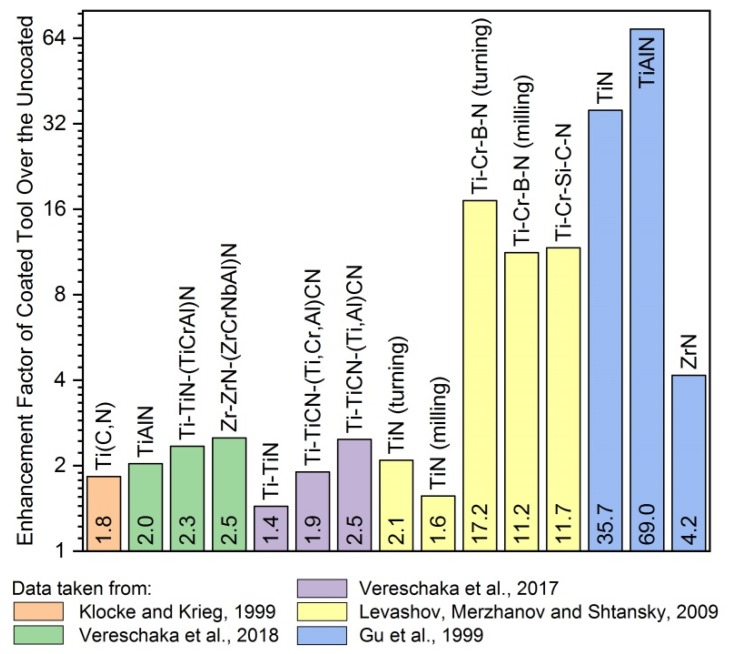
Comparison of enhancement factors of coated tools over uncoated tools (based on their lifetime). Analysed data are taken from selected published works [94,95,96,97,98]. The results strongly depend on operational conditions, such as the cutting method and speed, the workpiece material, and the thickness of the protective coating.

**Figure 8 materials-13-01377-f008:**
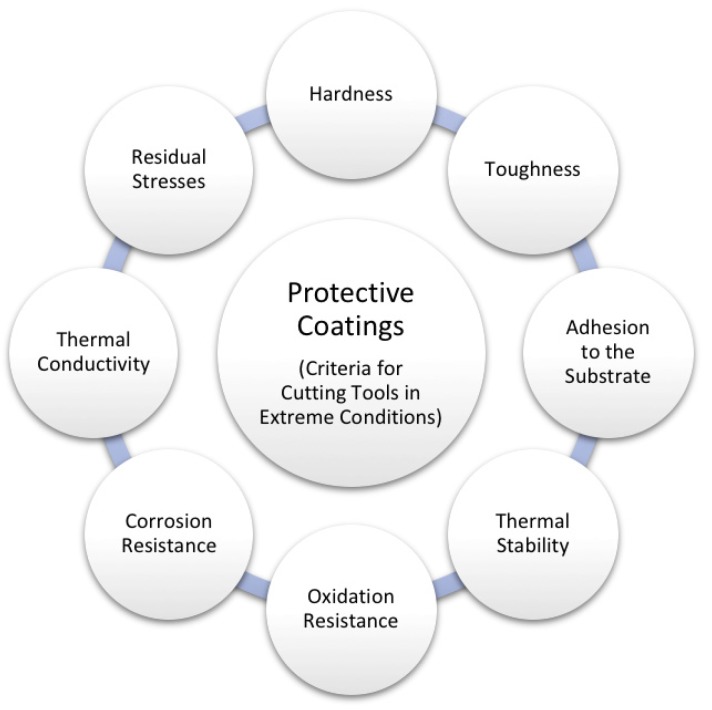
Criteria required for the success of the protective coatings of cutting tools.

**Figure 9 materials-13-01377-f009:**
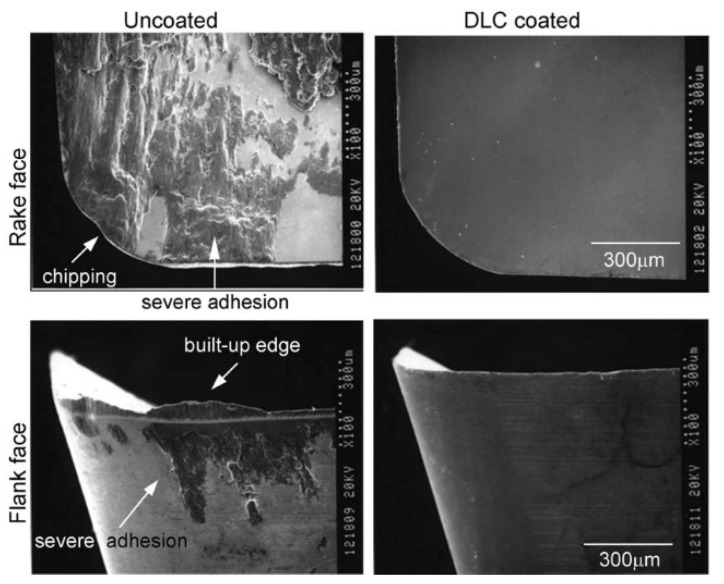
Scanning electron microscope (SEM) micrographs of the rake and flank face after the dry milling test for an AlCu_2.5_Si_18_ alloy [109].

**Figure 10 materials-13-01377-f010:**
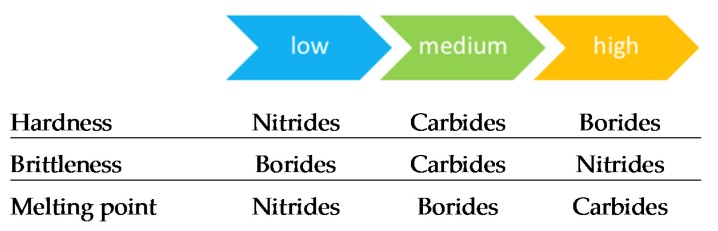
Properties of transition metal nitrides, carbides and borides [117].

**Figure 11 materials-13-01377-f011:**
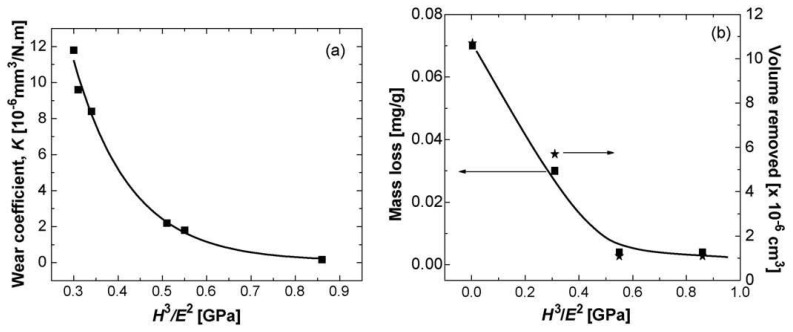
Tribological characteristics of TiN and nanocomposite nc-TiN/SiN_1.3_ films as a function of the H^3^/E^2^ ratio: (**a**) wear coefficient; (**b**) erosion rate [132].

**Figure 12 materials-13-01377-f012:**
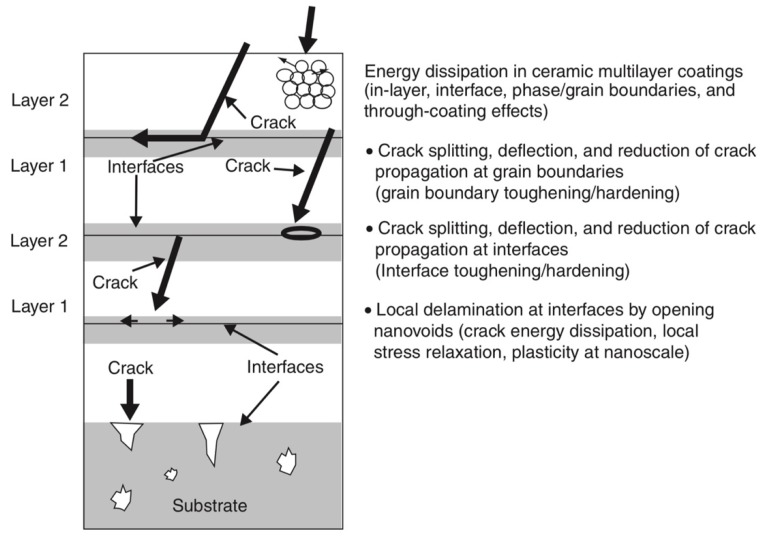
Toughening and strengthening mechanisms in multilayer coatings (taken from [158]).

**Figure 13 materials-13-01377-f013:**
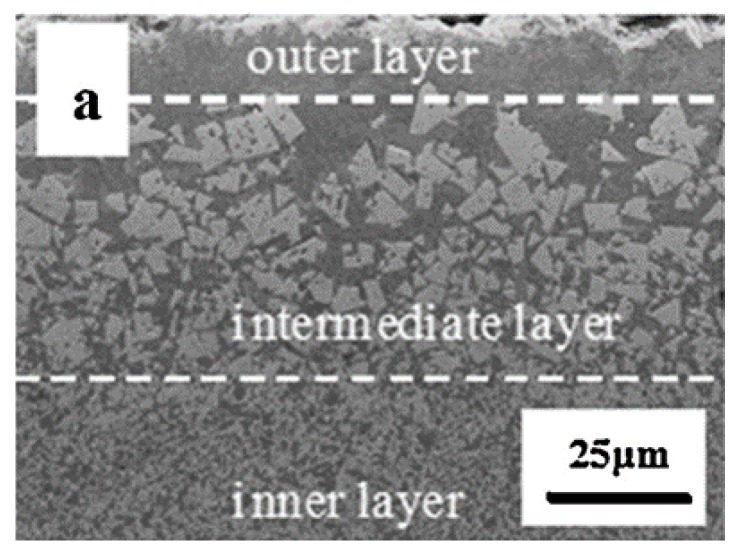
Cross-section SEM image of the surface of a graded cemented carbide material obtained in [201].

**Figure 14 materials-13-01377-f014:**
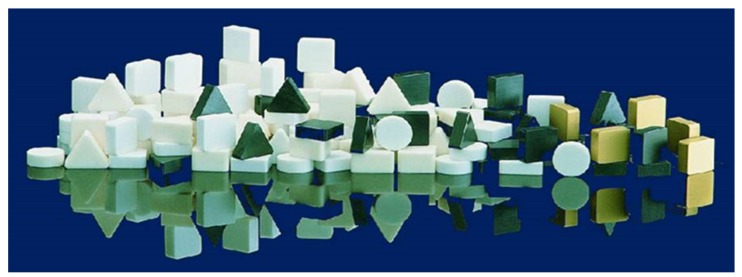
Tool inserts made of ceramics.

**Figure 15 materials-13-01377-f015:**
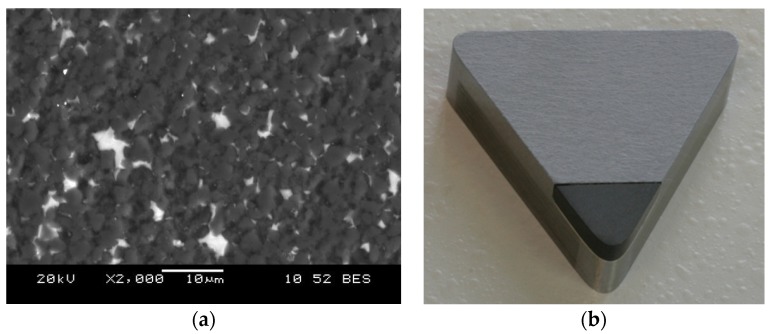
(**a**) Microstructure of the samples for a diamond composite with 5 wt.% TiB_2_ and (**b**) an insert with a diamond cutting edge.

**Figure 16 materials-13-01377-f016:**
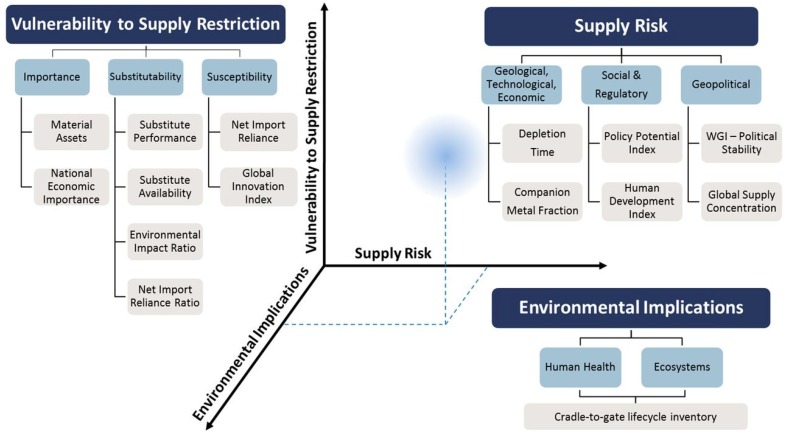
The methodology of criticality, showing the importance of substitution in the vertical axis (see [259]).

**Figure 17 materials-13-01377-f017:**
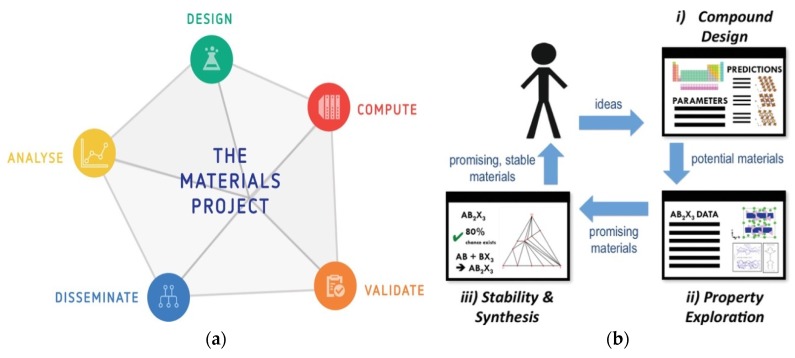
(**a**) Materials project thrusts. (**b**) In silico prototyping and iterative design steps of materials [292].

**Figure 18 materials-13-01377-f018:**
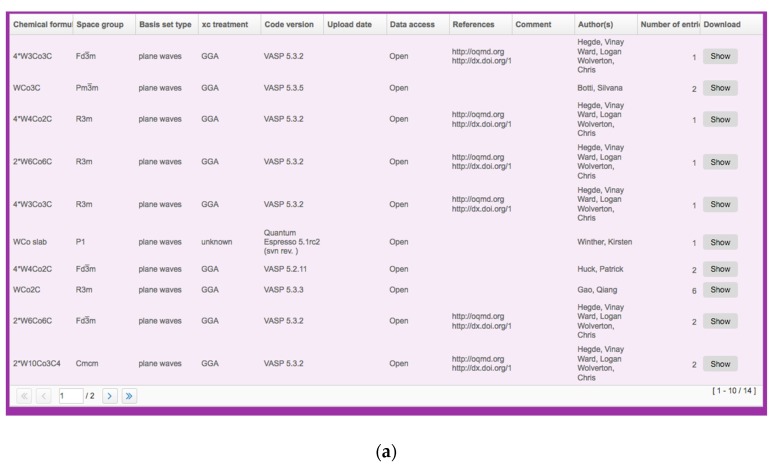
Search results for W–C–Co in two state-of-the-art computational data repositories. (**a**) In NOMAD [266]. (**b**) In the Materials Project [275].

**Figure 19 materials-13-01377-f019:**
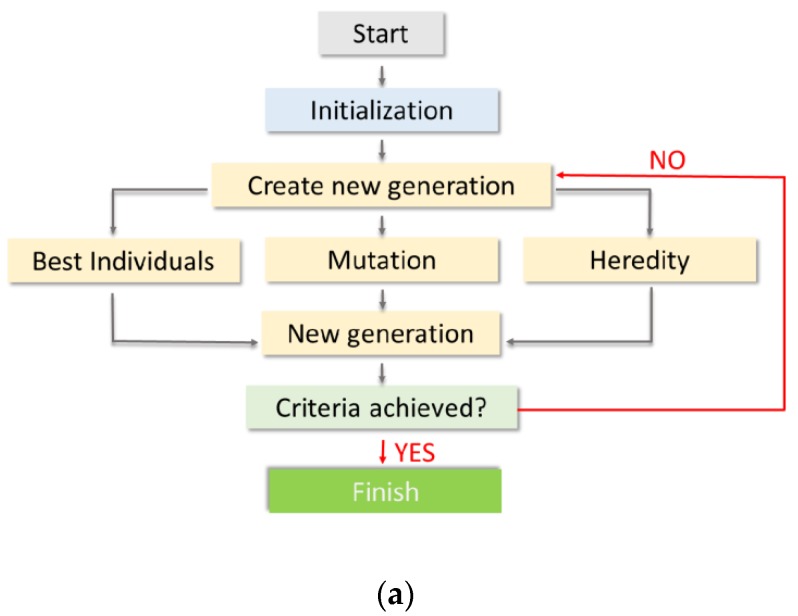
(**a**) Flowchart of a genetic algorithm as USPEX [309]. (**b**) Ashby plot of predicted new W-B phases (red points) compared to known superhard materials (blue points) [308].

**Figure 20 materials-13-01377-f020:**
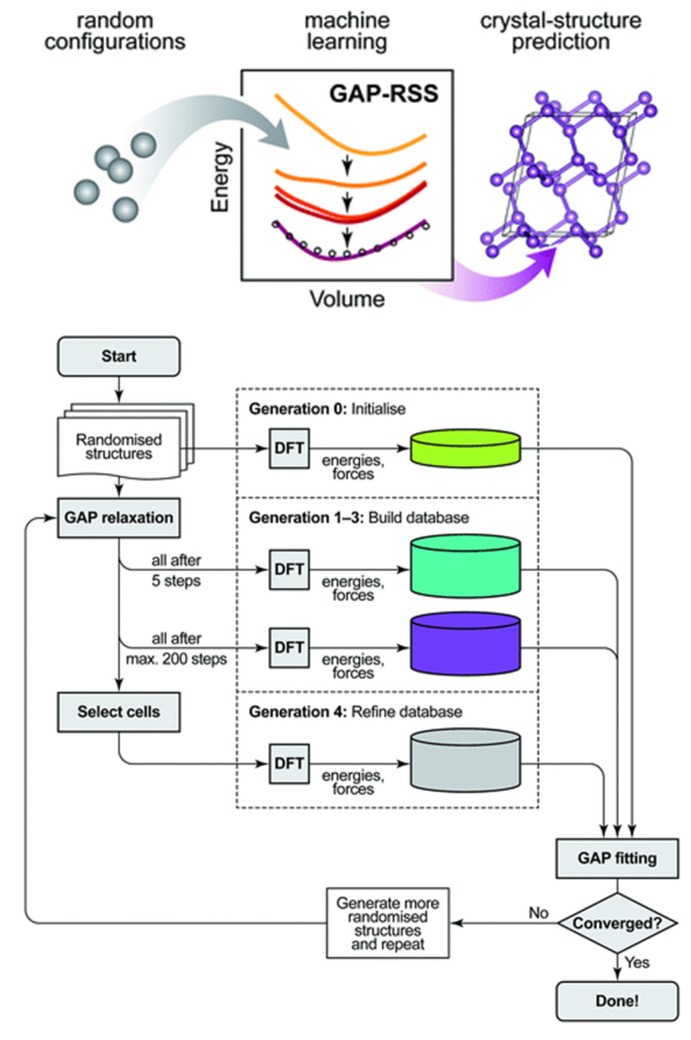
GAP-RSS (Gaussian approximation potential-based random structure searching) scheme and protocol [314].

**Table 1 materials-13-01377-t001:** Tabulated summary of recent efforts made in the vibration-assisted machining (VAM) of titanium alloys and steel.

Work Material	Cutting Parameters Used	Oscillation Parameters (Frequency (f), Amplitude (a))	Cutting Force Comparison with Conventional Turning	Additional Conclusions
Ti6Al2Sn4Zr6Mo (α + β Ti alloy) [69]	fr = 0.1 mm/rev; v = 10–60 m/min; d = 0.2 mm	f = 20 kHz; a = 10 µm	Reduction by 74%	Surface roughness improved by 50%
Ti-15333 (β alloy) [70]	fr = 100 µm/rev; v = 10 m/min; d = 100–500 µm	F = 20 kHz a = 8 µm	Reduction by 80%–85%	Surface roughness improved by 50% while heat was applied during ultrasonic assisted machining
Ti6Al4V [71]	fr = 0.1 mm/rev; v = 10–300 m/min d = 0.1mm	f = 20 kHz a = 20 µm	Reduction by 40%–45%	Surface roughness improved by 40%
Ti 15-3-3-3 (β Ti-alloy) [72]	Fr = 0.1 mm/rev; v = 10–70 m/min; d = 50–500 µm	f = 17.9 kHz a = 10 µm	Reduction by 71%–88%	Surface roughness improved by 49%
Low alloy steel (DF2) [73]	Fr = 0.1 mm/rev; v = 50 m/min; d = 0.2 mm	F = 19 kHz; a = 15 µm	Reduction by 50%	Tool wear 20% less

**Table 2 materials-13-01377-t002:** Modified form of measures suggested for improved machinability [37].

S.No.	Theoretical Approach	Experimental Realization
Modification of the process		
1	Reduction of chemical reaction rate between the tool and workpiece	Cryogenic turning [79]
2	Reduction of contact time between tool and workpiece	Vibration-assisted cutting [68,69,73,80,81,82,83,84,85,86]
3	Lowering of temperature rise and chemical contact	Usage of appropriate coolant [87,88]
Modification of the cutting tool		
4	Building a diffusion barrier on the cutting tool	Use of protective coatings [89]
5	Modifying the cutting tool geometry	Providing nanogrooves on the cutting tool
6	Use of alternative cutting tool material	Use of cBN
Workpiece modification		
7	Surface layer modification of the workpiece prior to cutting	Ion implantation
8	Surface defect machining	Pre-drilled laser holes in the workpiece reduce the shear strength, which is evident by observing a lower shear plane angle [90,91,92]

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
