# Peer review of "The Critical Raw Materials in Cutting Tools for Machining Applications: A Review"

_materials, 2020, doi:10.3390/ma13061377_

Round 1
Reviewer 1 Report
The manuscript is in good shape but please see my suggestions for the publication of this work.
I believe title of the work can be improved and made appealing for audience. Page 4 (line 23) – Multi materials and high entropy coatings are mentioned. Please explain more. Please check the citation in figure 10 captions – There is a mistake. [97 97]. In section 3 a lot of different cutting tool material coatings are discussed. However, it would be interesting to see if you can add briefly the associated wear mechanisms with respect to the evidence. In modeling and simulation section, there is no information about finite element (FE) assisted computation. The paper discusses some good sections and sub-sections, but when conclusions are drawn, I am expecting more detailed conclusions and also clear futuristic directions from each subsection for other researchers in the metal cutting community.
Author Response
Reviewer 1 |
|
I believe title of the work can be improved and made appealing for audience. |
We thank you for your suggestion. The new title is “The critical raw materials in cutting tools for machining applications: a review” |
Page 4 (line 23) – Multi materials and high entropy coatings are mentioned. Please explain more.
|
In the introduction, HEAs and multimaterials coatings are only mentioned. Later in the text, they are more extensively described. In the sentence at page 4, the authors anticipate what explained later in the text, i.e. that multi-elemental compounds and HEAs contain combinations of 5 or more elements and such complex combinations often includes CRMs as just a probability factor and(or) due to some required and specific properties. This is also mentioned in the next sentence of the paper and in Fig.2. Reference 12 reports about high entropy nitride/carbides/oxides forming elements such as Al, Ti, Cr, Si, Nb, Zr, etc, among which Si and Nb are critical. We believe that additional explanation are not needed here. |
Please check the citation in figure 10 captions – There is a mistake. [97 97]. |
Thank you, the citation was fixed.
|
In section 3 a lot of different cutting tool material coatings are discussed. However, it would be interesting to see if you can add briefly the associated wear mechanisms with respect to the evidence.
|
We thank the Reviewer for his suggestion. However, wear in cutting tools, due to the variety of tool materials, operating conditions and protective coatings, would require a review itself. This goes beyond the purpose of the present review where the focus is on critical raw materials. As a general assertion, in the text it is said “It is seen that coated cutting tools may have an extra lifetime of 200-500 % and more at the same cutting velocities. This also can lead to the increase of operational velocities (by 50 to 150 %) for the same lifetime of cutting tools”. These affirmations are also supported by data in Figure 7, showing the comparison of enhancement factors of coated tools over the uncoated ones. Some wear values of coated and uncoated tools is reported in the different sections, separately. |
In modeling and simulation section, there is no information about finite element (FE) assisted computation.
|
This was initially left out of the scope of this particular section, considering that there is neither that much information on other lower time and length scale modelling approaches, except for the comments in the “machine learning” subsection on the applications to kinetic Monte Carlo, molecular dynamics and DFT techniques, which are intended to be nothing but simple examples. However, we have included a short sentence and the corresponding references in the text, that now reads as follows: The sentence in section 4.3 originally starting with “…or DFT-KMC multiscale modelling approaches [298-299]…” was modified as “…or DFT-KMC [320-321] and finite element [322-324] multiscale modelling approaches are paradigmatic examples of advanced materials simulation methodologies that make use of machine learning–related techniques” References 320-324 were changed accordingly: [320] C. González, M. Panizo-Laiz, N. Gordillo, C.L. Guerrero, E. Tejado, F. Munnik, P. Piaggi, E. Bringa, R. Iglesias, J.M. Perlado, R. González-Arrabal, H trapping and mobility in nanostructured tungsten grain boundaries: a combined experimental and theoretical approach, Nucl. Fusion. 55 (2015) 113009. doi:10.1088/0029-5515/55/11/113009. [321] G. Valles, M. Panizo-Laiz, C. González, I. Martin-Bragado, R. González-Arrabal, N. Gordillo, R. Iglesias, C.L. Guerrero, J.M. Perlado, A. Rivera, Influence of grain boundaries on the radiation-induced defects and hydrogen in nanostructured and coarse-grained tungsten, Acta Mater. 122 (2017) 277–286. doi:10.1016/j.actamat.2016.10.007. [322] The Minerals, Metals & Materials Society (TMS), Advanced Computation and Data in Materials and Manufacturing: Core Knowledge Gaps and Opportunities (Pittsburgh, PA: TMS, 2018), doi: 10.7449/coreknowledge_1 [323] Bessa, M. A., Glowacki, P., Houlder, M., Bayesian Machine Learning in Metamaterial Design: Fragile Becomes Supercompressible. Adv. Mater. 2019, 31, 1904845, doi: 10.1002/adma.201904845. [324] White paper on gaps and obstacles in materials modelling , Kersti Hermansson, Pietro Asinari, Roy Chantrell, et al. eds. (EMMC-CSA 2019), https://emmc.info
|
The paper discusses some good sections and sub-sections, but when conclusions are drawn, I am expecting more detailed conclusions and also clear futuristic directions from each subsection for other researchers in the metal cutting community. |
We thank the Reviewer for his suggestion. The authors changed the structure of the conclusions trying to underline the proposed methodologies. |
Reviewer 2 Report
Review paper:
Past, present and future of the critical raw materials in cutting tools for machining applications
Reviewer comments
In this paper, authors reviewed published literature on different aspects to handle the issues related to critical raw materials (CRM) used in metal cutting tools. The main objective is to raise awareness and developing control strategies to reduce the use of CRMs. Based on EU Raw Materials Initiative and targeted parallel actions, efforts made to establish the state-of-the-art for use of CRMs in metal cutting.
According to authors, the cemented carbides capture one half of the market followed by high speed steel. Ceramics, cermets and superhard materials like polycrystalline diamond (PCD) and polycrystalline cubic boron nitride (PCBN) capture the remaining share of the tool material market. The two important ingredients of the carbide tools, namely, tungsten and cobalt, were identified (in 2011) among the list of the 14 critical raw materials vital to EU industries.
The review is timely and comprehensive, but very general in nature. The issues were raised very nicely identifying W and Co as the major CRMs used in the mostly commonly tool material (WC). But, the literature and discussions are very general and lack in highlighting their importance in the context of specific material WC. Title itself is confusing as nothing reviewed in terms of past, present and future of the critical raw materials in cutting tools. Section 2 is on attempts to expand the life of WC-Co based tools. But, general description and advantages of advanced machining techniques and coating are presented. No relationship established with WC-Co based tools. Although relevant, it was expected that authors will review the efforts made and reported in the published literature for WC or tools materials having CRMs. The results on the cryogenic treatment of tungsten carbide cutting tools were summarised at line 257. But, no reference is cited for summarised points. Some of them were not even discussed in the preceding paragraph. In addition, the findings of uncoated and coated WC insets were mixed. The results and discussion should focus on the uncoated WC inserts. Role of coating should be discussed separately. Failed to understand why only cBN in the table II. There are so many other alternate tools are being investigated and reported. In addition, no reference number is given. Figure 6 gives only the distribution in percent of uncoated and coated WC tools. It is not clear how the demand changed over the timeline. In Figure 7 different referencing is used. Be consistent and use number only such as [73] instead of Klocke and Krieg, 1999. Fig 8 and related discussion is very general in nature. It should be specific to WC or other CRM. Mark (a) and (b) properly in Figure 4. The paper in its current form do not satisfy the purpose of a review paper. It was expected to be focused on CRMs such as W and Co in WC tools. Therefore, not recommended for publication. It is basically a collection of many papers on several aspects of machining having no depth and focus.Author Response
Reviewer 2 |
|
The review is timely and comprehensive, but very general in nature. The issues were raised very nicely identifying W and Co as the major CRMs used in the mostly commonly tool material (WC). But, the literature and discussions are very general and lack in highlighting their importance in the context of specific material WC. |
We thank Reviewer for his suggestion. We have tried to focus and emphasize W and Co importance in the context of specific WC material by inserting comments and sentences in almost all paragraphs. |
Title itself is confusing as nothing reviewed in terms of past, present and future of the critical raw materials in cutting tools. |
We thank the Reviewer for his suggestion. The new title is “The critical raw materials in cutting tools for machining applications: a review”. |
Section 2 is on attempts to expand the life of WC-Co based tools. But, general description and advantages of advanced machining techniques and coating are presented. No relationship established with WC-Co based tools. Although relevant, it was expected that authors will review the efforts made and reported in the published literature for WC or tools materials having CRMs. |
Following the reviewer’s comment, a subsection dedicated to WC-Co tools was added (par. 2. 1) to better focus the description on this aspect.
Section 2.3 (Protective coatings) describes the main classes of hard coatings used to protect and increase lifetime of WC-Co cutting tools. Even though the description may seem general, the described coatings are generally applied to WC-Co and other tools materials, as well. Regarding the PVD and CVD nitride coatings the text already specified it (“The first industrial CVD coated tool coating on cemented carbide was TiC in 1969, and in 1980 TiN became the first PVD coating”). Reference to WC-Co tools is also reported in the description of diamond-like coatings. However, to more highlight the link to WC-Co tools, as suggested by the Reviewer, the following sentence was added before 2.3.1: “In the next section a brief description of the main properties of the hard coatings mainly used to coat WC-Co and other tool materials is described.” Also, some sentences have been added throughout par. 2.3 to better evidence the application of the described coatings specifically on WC-Co tools or tools containing other CRMs. |
The results on the cryogenic treatment of tungsten carbide cutting tools were summarised at line 257. But, no reference is cited for summarised points. Some of them were not even discussed in the preceding paragraph. In addition, the findings of uncoated and coated WC insets were mixed. The results and discussion should focus on the uncoated WC inserts. Role of coating should be discussed separately. |
The authors included a reference by Chopra and Sargade [62] which is an overview of the metallurgy underlying the cryogenic treatment of cutting tools with reference also to the results related to WC-Co tools. The scientific community is still discussing the role of coatings in cryogenic machining. It seemed appropriate to add also a very recent reference of Biswal et al. [66] in which it is highlighted how uncoated tools perform better in cryogenic conditions. |
Failed to understand why only cBN in the table II. There are so many other alternate tools are being investigated and reported. In addition, no reference number is given. |
CBN was stated as an example in place of diamond tool. It is true that many tools are being investigated but the use of tool is governed by the material to cut and the precision required. Diamond tool for instance achieved highest possible precision and except for cBN as an alternative no other tool materials can rival. |
Figure 6 gives only the distribution in percent of uncoated and coated WC tools. It is not clear how the demand changed over the timeline. |
Authors are very grateful for the comment which they found valuable, but unfortunately authors do not have access to the absolute values, which are probably stated in not public reports available by pre-paid subscription or only for commercial sector. However, the idea of the figure was to demonstrate the dynamic of the coated tools dominance through the years.
The reference in the caption is fixed - [7] (Bobzin) instead of [71]. And "Data from" instead of "data from" (capital letter). |
In Figure 7 different referencing is used. Be consistent and use number only such as [73] instead of Klocke and Krieg, 1999. |
Authors used different reference system in Fig. 7 with the purpose that this figure later could be directly republished or used in other papers/presentations with the reference to our review-paper. It improves significantly this process. Additionally, the caption contains the references in standard mode through the manuscript with all details in the reference list. |
Fig 8 and related discussion is very general in nature. It should be specific to WC or other CRM. |
The idea of the figure is to show general principles and requirements for protective coatings, which then could be applied for any workpiece materials, including WC or CRM-based. |
Mark (a) and (b) properly in Figure 4. |
(a) and (b) in Figure 4 were properly marked |
The paper in its current form do not satisfy the purpose of a review paper. It was expected to be focused on CRMs such as W and Co in WC tools. Therefore, not recommended for publication. It is basically a collection of many papers on several aspects of machining having no depth and focus. |
The authors believe that the paper meets the standards of the journal in this revised form and could be of a high interest of potential readers. |
Reviewer 3 Report
The paper presents an intensive review of the litterature in order to make the state of the art of advanced techiques allowing to reduce the use of critical raw material for machining applications. The paper is well written and analysis more thant three hundred relevant references. Only some minor comments can be made:
the interest of the equations in figure 4 (c) is questionable, maybe it can be removed par 2.1.2 is entitled cryogenic machining, however it mixes actual cryogenic machining and cryo-treated tool material. It may be clearer to separate those two aspects vertical axis of figure 7 is not clear, the enhancement factor (seems to be based on toollife) must be defines clearly in section 2.2.4, it is stated that the ratio E/H^2 plays an important role, but the graphs in figure11 shows H^3/E^2 value, it seems odd
Author Response
Reviewer 3 |
|
the interest of the equations in figure 4 (c) is questionable, maybe it can be removed
|
We thank the Reviewer for his suggestion. We removed the equations in Figure 4 which is now easily readable. |
par 2.1.2 is entitled cryogenic machining, however it mixes actual cryogenic machining and cryo-treated tool material. It may be clearer to separate those two aspects |
This document was born from the COST Action CA15102 “Solutions for Critical Raw Materials under Extreme Conditions”, focused on raising awareness to inspire practices that reduce dependence on CRMs. “2.2.2 Cryogenic machining” does not claim to be an exhaustive section on cryogenics in its various aspects, but would like to emphasize how cryogenics can also contribute to saving critical materials by increasing life time. Nevertheless, two sentences and two new and recent references have been introduced.
|
vertical axis of figure 7 is not clear, the enhancement factor (seems to be based on toollife) must be defines clearly |
The caption of the figure was corrected according to your suggestion. |
in section 2.2.4, it is stated that the ratio E/H^2 plays an important role, but the graphs in figure11 shows H^3/E^2 value, it seems odd |
The typo-error in the text was fixed, and H3/E2 was substituted to "H/E” |
Reviewer 4 Report
This paper provides a timely curated review of related research and focuses on raising awareness to inspire practices that reduce the reliance on CRMs. While this paper covers technical details of work where I am not an expert, it seems to be a good contribution and is well written. I provide the following suggestions to improve the manuscript:
1-Figure 1 is pixelated and could be made more clear
2-Figure 2 could benefit from a cited source
3-Figure 3 resolution is not so good--a more clear image can be used
4-Figure 4 from line 166, the image spills over to the next page, but it seems rather odd to have 2 images and a table that are all one figure, perhaps split this to be more readable.
5-Figure 15 should have an A and B?
Aside from these suggestions, the paper seems to be a good contribution providing a good overview with a focus on materials, prevalent and new industrial techniques and sustainability.
Author Response
Reviewer 4 |
|
This paper provides a timely curated review of related research and focuses on raising awareness to inspire practices that reduce the reliance on CRMs. While this paper covers technical details of work where I am not an expert, it seems to be a good contribution and is well written. I provide the following suggestions to improve the manuscript: |
|
1-Figure 1 is pixelated and could be made more clear |
We thank the reviewer for his/her suggestions. Figure 1 has been replaced with a higher resolution one. |
2-Figure 2 could benefit from a cited source |
Figure 2 reports the periodic table with highlighted the elements included in the CRMs lists released by EC in 2011, 2014, 2017. It was made by the authors. We cited, however, in the caption the sources of CRMs list. |
3-Figure 3 resolution is not so good--a more clear image can be used |
This is a correctly cited figure of the article whose features faithfully reproduce the original resolution.
|
4-Figure 4 from line 166, the image spills over to the next page, but it seems rather odd to have 2 images and a table that are all one figure, perhaps split this to be more readable. |
We removed the equations in Figure 4 which is now more readable. |
5-Figure 15 should have an A and B? |
We introduced a and b in figure 15. |
Aside from these suggestions, the paper seems to be a good contribution providing a good overview with a focus on materials, prevalent and new industrial techniques and sustainability.
|
|
Round 2
Reviewer 1 Report
-
Reviewer 2 Report
Accept as authors tried to incorporate all previous comments.